# ENERGY-BASED TEST SAMPLE ADAPTATION FOR DOMAIN GENERALIZATION

**Zehao Xiao[1], Xiantong Zhen[1,2]\*, Shengcai Liao[2], Cees G. M. Snoek[1]**
[1]AIM Lab, University of Amsterdam  [2]Inception Institute of Artificial Intelligence

## ABSTRACT

In this paper, we propose energy-based sample adaptation at test time for domain generalization. Where previous works adapt their models to target domains, we adapt the unseen target samples to source-trained models. To this end, we design a discriminative energy-based model, which is trained on source domains to jointly model the conditional distribution for classification and data distribution for sample adaptation. The model is optimized to simultaneously learn a classifier and an energy function. To adapt target samples to source distributions, we iteratively update the samples by energy minimization with stochastic gradient Langevin dynamics. Moreover, to preserve the categorical information in the sample during adaptation, we introduce a categorical latent variable into the energy-based model. The latent variable is learned from the original sample before adaptation by variational inference and fixed as a condition to guide the sample update. Experiments on six benchmarks for classification of images and microblog threads demonstrate the effectiveness of our proposal.

## 1 INTRODUCTION

Deep neural networks are vulnerable to domain shifts and suffer from lack of generalization on test samples that do not resemble the ones in the training distribution (Recht et al., 2019; Zhou et al., 2021; Krueger et al., 2021; Shen et al., 2022). To deal with the domain shifts, domain generalization has been proposed (Muandet et al., 2013; Gulrajani & Lopez-Paz, 2020; Cha et al., 2021). Domain generalization strives to learn a model exclusively on source domains in order to generalize well on unseen target domains. The major challenge stems from the large domain shifts and the unavailability of any target domain data during training.

To address the problem, domain invariant learning has been widely studied, e.g., (Motiian et al., 2017; Zhao et al., 2020; Nguyen et al., 2021), based on the assumption that invariant representations obtained on source domains are also valid for unseen target domains. However, since the target data is inaccessible during training, it is likely an "adaptivity gap" (Dubey et al., 2021) exists between representations from the source and target domains. Therefore, recent works try to adapt the classification model with target samples at *test time* by further fine-tuning model parameters (Sun et al., 2020; Wang et al., 2021) or by introducing an extra network module for adaptation (Dubey et al., 2021). Rather than adapting the model to target domains, Xiao et al. (2022) adapt the classifier for each sample at test time. Nevertheless, a single sample would not be able to adjust the whole model due to the large number of model parameters and the limited information contained in the sample. This makes it challenging for their method to handle large domain gaps. Instead, we propose to adapt each target sample to the source distributions, which does not require any fine-tuning or parameter updates of the source model.

In this paper, we propose energy-based test sample adaptation for domain generalization. The method is motivated by the fact that energy-based models (Hinton, 2002; LeCun et al., 2006) flexibly model complex data distributions and allow for efficient sampling from the modeled distribution by Langevin dynamics (Du & Mordatch, 2019; Welling & Teh, 2011). Specifically, we define a new discriminative energy-based model as the composition of a classifier and a neural-network-based energy function in the data space, which are trained simultaneously on the source domains. The trained model iteratively

---

\*Currently with United Imaging Healthcare, Co., Ltd., China.

updates the representation of each target sample by gradient descent of energy minimization through Langevin dynamics, which eventually adapts the sample to the source data distribution. The adapted target samples are then predicted by the classifier that is simultaneously trained in the discriminative energy-based model. For both efficient energy minimization and classification, we deploy the energy functions on the input feature space rather than the raw images.

Since Langevin dynamics tends to draw samples randomly from the distribution modeled by the energy function, it cannot guarantee category equivalence. To maintain the category information of the target samples during adaptation and promote better classification performance, we further introduce a categorical latent variable in our energy-based model. Our model learns the latent variable to explicitly carry categorical information by variational inference in the classification model. We utilize the latent variable as conditional categorical attributes like in compositional generation (Du et al., 2020a; Nie et al., 2021) to guide the sample adaptation to preserve the categorical information of the original sample. At inference time, we simply ensemble the predictions obtained by adapting the unseen target sample to each source domain as the final domain generalization result.

We conduct experiments on six benchmarks for classification of images and microblog threads to demonstrate the promise and effectiveness of our method for domain generalization[1].

## 2 METHODOLOGY

In domain generalization, we are provided source and target domains as non-overlapping distributions on the joint space $\mathcal{X} \times \mathcal{Y}$, where $\mathcal{X}$ and $\mathcal{Y}$ denote the input and label space, respectively. Given a dataset with $S$ source domains $\mathcal{D}_s = \left\{ D_s^i \right\}_{i=1}^{S}$ and $T$ target domains $\mathcal{D}_t = \left\{ D_t^i \right\}_{i=1}^{T}$, a model is trained only on $\mathcal{D}_s$ and required to generalize well on $\mathcal{D}_t$. Following the multi-source domain generalization setting (Li et al., 2017; Zhou et al., 2021), we assume there are multiple source domains with the same label space to mimic good domain shifts during training.

In this work, we propose energy-based test sample adaptation, which adapts target samples to source distributions to tackle the domain gap between target and source data. The rationale behind our model is that adapting the target samples to the source data distributions is able to improve the prediction of the target data with source models by reducing the domain shifts, as shown in Figure 1 (left). Since the target data is never seen during training, we mimic domain shifts during the training stage to learn the sample adaptation procedure. By doing so, the model acquires the ability to adapt each target sample to the source distribution at inference time. In this section, we first provide a preliminary on energy-based models and then present our energy-based test sample adaptation.

### 2.1 ENERGY-BASED MODEL PRELIMINARY

Energy-based models (LeCun et al., 2006) represent any probability distribution $p(\mathbf{x})$ for $\mathbf{x} \in \mathbb{R}^D$ as $p_\theta(\mathbf{x}) = \frac{\exp(-E_\theta(\mathbf{x}))}{Z_\theta}$, where $E_\theta(\mathbf{x}) : \mathbb{R}^D \to \mathbb{R}$ is known as the energy function that maps each input sample to a scalar and $Z_\theta = \int \exp(-E_\theta(\mathbf{x}))d\mathbf{x}$ denotes the partition function. However, $Z_\theta$ is usually intractable since it computes the integration over the entire input space of $\mathbf{x}$. Thus, we cannot train the parameter $\theta$ of the energy-based model by directly maximizing the log-likelihood $\log p_\theta(\mathbf{x}) = -E_\theta(x) - \log Z_\theta$. Nevertheless, the log-likelihood has the derivative (Du & Mordatch, 2019; Song & Kingma, 2021) :

$$\frac{\partial \log p_\theta(\mathbf{x})}{\partial \theta} = \mathbb{E}_{p_d(\mathbf{x})}\left[ -\frac{\partial E_\theta(\mathbf{x})}{\partial \theta} \right] + \mathbb{E}_{p_\theta(\mathbf{x})}\left[ \frac{\partial E_\theta(\mathbf{x})}{\partial \theta} \right], \tag{1}$$

where the first expectation term is taken over the data distribution $p_d(\mathbf{x})$ and the second one is over the model distribution $p_\theta(\mathbf{x})$.

The objective function in eq. (1) encourages the model to assign low energy to the sample from the real data distribution while assigning high energy to those from the model distribution. To do so, we need to draw samples from $p_\theta(\mathbf{x})$, which is challenging and usually approximated by MCMC methods (Hinton, 2002). An effective MCMC method used in recent works (Du & Mordatch, 2019; Nijkamp et al., 2019; Xiao et al., 2021b; Grathwohl et al., 2020) is Stochastic Gradient Langevin

---

[1] Code available: https://github.com/zzzx1224/EBTSA-ICLR2023.

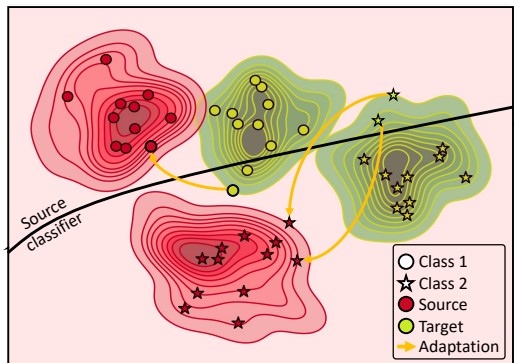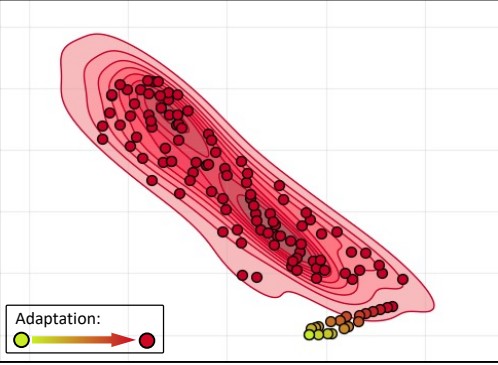

Figure 1: **Illustration of the proposed energy-based model.** It aims to adapt the target samples to the source distributions, which can be more accurately classified by the source domain classifier (left). With Langevin dynamics, each target sample is adapted iteratively to the source data distributions, which are represented as the gradient colors from green to red (right). Best viewed in color.

Dynamics (Welling & Teh, 2011), which simulates samples by

$$\mathbf{x}^{i+1} = \mathbf{x}^i - \frac{\lambda}{2}\frac{\partial E_\theta(\mathbf{x}^i)}{\partial \mathbf{x}^i} + \epsilon, \quad s.t., \quad \epsilon \sim \mathcal{N}(0, \lambda) \tag{2}$$

where $\lambda$ denotes the step-size and $\mathbf{x}^0$ is drawn from the initial distribution $p_0(\mathbf{x})$, which is usually a uniform distribution (Du & Mordatch, 2019; Grathwohl et al., 2020).

Actually, maximizing $\log p_\theta(\mathbf{x})$ is equivalent to minimizing the KL divergence $\mathbb{D}_{\mathrm{KL}}(p_d(\mathbf{x})||p_\theta(\mathbf{x}))$ (Song & Kingma, 2021), which is alternatively achieved in (Hinton, 2002) by minimizing contrastive divergence:

$$\mathbb{D}_{\mathrm{KL}}(p_d(\mathbf{x})||p_\theta(\mathbf{x})) - \mathbb{D}_{\mathrm{KL}}(q_\theta(\mathbf{x})||p_\theta(\mathbf{x})), \tag{3}$$

where $q_\theta(\mathbf{x}) = \prod_\theta^t p_d(\mathbf{x})$, representing $t$ sequential MCMC transitions starting from $p(\mathbf{x})$ (Du et al., 2021a) and minimizing eq. (3) is achieved by minimizing:

$$\mathbb{E}_{p_d(\mathbf{x})}[E_\theta(\mathbf{x})] - \mathbb{E}_{\mathrm{stop\_grad(q_\theta(\mathbf{x}))}}[E_\theta(\mathbf{x})] + \mathbb{E}_{q_\theta(\mathbf{x})}[E_{\mathrm{stop\_grad}(\theta)}(\mathbf{x})] + \mathbb{E}_{q_\theta(\mathbf{x})}[\log q_\theta(\mathbf{x})]. \tag{4}$$

Eq. (4) avoids drawing samples from the model distribution $p_\theta(\mathbf{x})$, which often requires an exponentially long time for MCMC sampling (Du et al., 2021a). Intuitively, $q_\theta(\mathbf{x})$ is closer to $p_\theta(\mathbf{x})$ than $p_d(\mathbf{x})$, which guarantees that $\mathbb{D}_{\mathrm{KL}}(p_d(\mathbf{x})||p_\theta(\mathbf{x})) \geq \mathbb{D}_{\mathrm{KL}}(q_\theta(\mathbf{x})||p_\theta(\mathbf{x}))$ and eq. (3) can only be zero when $p_\theta(\mathbf{x}) = p_d(\mathbf{x})$.

## 2.2 ENERGY-BASED TEST SAMPLE ADAPTATION

We propose energy-based test sample adaption to tackle the domain gap between source and target data distributions. This is inspired by the fact that Langevin dynamics simulates samples of the distribution expressed by the energy-based model through gradient-based updates, with no restriction on the sample initialization if the sampling steps are sufficient (Welling & Teh, 2011; Du & Mordatch, 2019). We leverage this property to conduct test sample adaptation with Langevin dynamics by setting the target sample as the initialization and updating it iteratively. With the energy-based model of the source data distribution, as shown in Figure 1 (right), target samples are gradually updated towards the source domain and with sufficient update steps, the target sample will eventually be adapted to the source distribution.

**Discriminative energy-based model.** We propose the discriminative energy-based model $p_{\theta,\phi}(\mathbf{x}, \mathbf{y})$ on the source domain, which is constructed by a classification model and an energy function in the data space. Note that $\mathbf{x}$ denotes the feature representations of the input image $I$, where $\mathbf{x}$ is generated by a neural network backbone $\mathbf{x}{=}f_\psi(I)$. Different from the regular energy-based models that generate data samples from uniform noise, our goal is to promote the discriminative task, i.e., the conditional distribution $p(\mathbf{y}|\mathbf{x})$, which is preferred to be jointly modeled with the feature distributions $p(\mathbf{x})$ of the input data. Thus, the proposed energy-based model is defined on the joint space of

$\mathcal{X} \times \mathcal{Y}$ to consider both the classification task and the energy function. Formally, the discriminative energy-based model of a source domain is formulated as:

$$p_{\theta,\phi}(\mathbf{x}, \mathbf{y}) = p_\phi(\mathbf{y}|\mathbf{x}) \frac{\exp(-E_\theta(\mathbf{x}))}{Z_{\theta,\phi}}, \tag{5}$$

where $p_\phi(\mathbf{y}|\mathbf{x})$ denotes the classification model and $E_\theta(\mathbf{x})$ is an energy function, which is nowadays implemented with neural networks. Eq. (5) enables the energy-based model to jointly model the feature distribution of input data and the conditional distribution on the source domains. An unseen target sample $\mathbf{x}_t$ is iteratively adapted to the distribution of the source domain $D_s$ by Langevin dynamics update with the energy function $E_\theta(\mathbf{x})$ and predicted by the classification model $p_\phi(\mathbf{y}|\mathbf{x})$.

The model parameters $\theta$ and $\phi$ can be jointly optimized following eq. (3) by minimizing:

$$\mathbb{D}_{\mathrm{KL}}(p_d(\mathbf{x}, \mathbf{y})||p_{\theta,\phi}(\mathbf{x}, \mathbf{y})) - \mathbb{D}_{\mathrm{KL}}(q_{\theta,\phi}(\mathbf{x}, \mathbf{y})||p_{\theta,\phi}(\mathbf{x}, \mathbf{y})), \tag{6}$$

which is derived as:

$$\begin{aligned}
\mathcal{L} = {} & -\mathbb{E}_{p_d(\mathbf{x},\mathbf{y})}\big[\log p_\phi(\mathbf{y}|\mathbf{x})\big] + \mathbb{E}_{p_d(\mathbf{x},\mathbf{y})}\big[E_\theta(\mathbf{x})\big] - \mathbb{E}_{\mathrm{stop\_grad}(q_{\theta,\phi}(\mathbf{x},\mathbf{y}))}\big[E_\theta(\mathbf{x})\big] \\
& + \mathbb{E}_{q_{\theta,\phi}(\mathbf{x},\mathbf{y})}\big[E_{\mathrm{stop\_grad}(\theta)}(\mathbf{x}) - \log p_{\mathrm{stop\_grad}(\phi)}(\mathbf{y}|\mathbf{x})\big],
\end{aligned} \tag{7}$$

where $p_d(\mathbf{x}, \mathbf{y})$ denotes the real data distribution and $q_{\theta,\phi}(\mathbf{x}, \mathbf{y}) = \prod_\theta^t p(\mathbf{x}, \mathbf{y})$ denotes $t$ sequential MCMC samplings from the distribution expressed by the energy-based model similar to eq. (4) (Du et al., 2021a). We provide the detailed derivation in Appendix A.

In eq. (7), the first term encourages to learn a discriminative classifier on the source domain. The second and third terms train the energy function to model the data distribution of the source domain by assigning low energy on the real samples and high energy on the samples from the model distribution. Different from the first three terms that directly supervise the model parameters $\theta$ and $\phi$, the last term stops the gradients of the energy function $E_\theta$ and classifier $\phi$ while back-propagating the gradients to the adapted samples $q_{\theta,\phi}(\mathbf{x}, \mathbf{y})$. Because of the stop-gradient, this term does not optimize the energy or log-likelihood of a given sample, but rather increases the probability of such samples with low energy and high log-likelihood under the modeled distribution.

Essentially, the last term trains the model $\theta$ to provide a variation for each sample that encourages its adapted version to be both discriminative on the source domain classifier and low energy on the energy function. Intuitively, it supervises the model to learn the ability to preserve categorical information during adaptation and find a faster way to minimize the energy.

**Label-preserving adaptation with categorical latent variable.** Since the ultimate goal is to correctly classify target domain samples, it is necessary to maintain the categorical information in the target sample during the iterative adaptation process. Eq. (7) contains a supervision term that encourages the adapted target samples to be discriminative for the source classification models. However, as the energy function $E_\theta$ operates only in the $\mathcal{X}$ space and the sampling process of Langevin dynamics tends to result in random samples from the sampled distribution that are independent of the starting point, there is no categorical information considered during the adaptation procedure.

To achieve label-preserving adaptation, we introduce a categorical latent variable $\mathbf{z}$ into the energy function to guide the adaptation of target samples to preserve the category information. With the latent variable, the energy function $E_\theta$ is defined in the joint space of $\mathcal{X} \times \mathcal{Z}$. The categorical information contained in $\mathbf{z}$ will be explicitly incorporated into the iterative adaptation. To do so, we define the energy-based model with the categorical latent variable as:

$$p_{\theta,\phi}(\mathbf{x}, \mathbf{y}) = \int p_{\theta,\phi}(\mathbf{x}, \mathbf{y}, \mathbf{z}) d\mathbf{z} = \int p_\phi(\mathbf{y}|\mathbf{z}, \mathbf{x}) p_\phi(\mathbf{z}|\mathbf{x}) \frac{\exp(-E_\theta(\mathbf{x}|\mathbf{z}))}{Z_{\theta,\phi}} d\mathbf{z}, \tag{8}$$

where $\phi$ denotes the parameters of the classification model that predicts $\mathbf{z}$ and $\mathbf{y}$ and $E_\theta$ denotes the energy function that models the distribution of $\mathbf{x}$ considering the information of latent variable $\mathbf{z}$. $\mathbf{z}$ is trained to contain sufficient categorical information of $\mathbf{x}$ and serves as the conditional attributes that guide the adapted samples $\mathbf{x}$ preserving the categorical information. Once obtained from the original input feature representations $\mathbf{x}$, $\mathbf{z}$ is fixed and taken as the input of the energy function together with the updated $\mathbf{x}$ in each iteration. Intuitively, when $\mathbf{x}$ is updated from the target domain to the source domain via Langevin dynamics, $\mathbf{z}$ helps it preserve the classification information contained in the original $\mathbf{x}$, without introducing additional information.

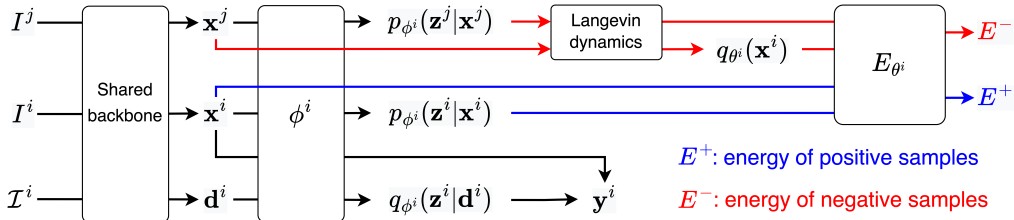

Figure 2: **Overall process of the proposed sample adaptation by discriminative energy-based model.** In each iteration, we train the classification model $\phi^i$ and energy function $E_{\theta^i}$ of one source domain $D^i$. $I^i$ and $I^j$ denote the images from domains $D^i$ and $D^j$, respectively. $\mathcal{I}^i$ denotes one batch of images, which generate the center features $\mathbf{d}^i$. The energy function $E_{\theta^i}$ is trained by using $\mathbf{x}^i$ as positive samples and adapted samples $q_{\theta^i}(\mathbf{x}^i)$ generated by $\mathbf{x}^j$ from other domains as negative samples. The adaptation is achieved by Langevin dynamics of $E_{\theta^i}$. During inference, the target samples are adapted by Langevin dynamics of the energy function $E_\theta$ of each source domain and then predicted by eq. (11).

To learn the latent variable $\mathbf{z}$ with more categorical information, we estimate $\mathbf{z}$ by variational inference and design a variational posterior $q_\phi(\mathbf{z}|\mathbf{d_x})$, where $\mathbf{d_x}$ is the average representation of samples from the same category as $\mathbf{x}$ on the source domain. Therefore, $q_\phi(\mathbf{z}|\mathbf{d_x})$ can be treated as a probabilistic prototypical representation of a class. By incorporating $q_\phi(\mathbf{z}|\mathbf{d_x})$ into eq. (8), we obtain the lower bound of the log-likelihood:

$$\log p_{\theta,\phi}(\mathbf{x},\mathbf{y}) \geq \mathbb{E}_{q_\phi}[\log p_\phi(\mathbf{y}|\mathbf{z},\mathbf{x}) - E_\theta(\mathbf{x},\mathbf{z}) - \log Z_{\theta,\phi}] + \mathbb{D}_{\mathrm{KL}}[q_\phi(\mathbf{z}|\mathbf{d_x})||p_\phi(\mathbf{z}|\mathbf{x})]. \quad (9)$$

Note that in eq. (9), the categorical latent variable $\mathbf{z}$ is incorporated into both the classification model $p_\phi(\mathbf{y}|\mathbf{z},\mathbf{x})$ and the energy function $E_\theta(\mathbf{x}|\mathbf{z})$. The energy function contains both data information and categorical information. During the Langevin dynamics update for sample adaptation, the latent variable provides categorical information in each iteration, which enables the adapted target samples to be discriminative.

By incorporating eq. (9) into eq. (6), we derive the objective function with the categorical latent variable as:

$$\begin{aligned}
\mathcal{L}_f = {} & \mathbb{E}_{p_d(\mathbf{x},\mathbf{y})}\Big[\mathbb{E}_{q_\phi(\mathbf{z})}[-\log p_\phi(\mathbf{y}|\mathbf{z},\mathbf{x})] + \mathbb{D}_{\mathrm{KL}}[q_\phi(\mathbf{z}|\mathbf{d_x})||p_\phi(\mathbf{z}|\mathbf{x})]\Big] + \mathbb{E}_{q_\phi(\mathbf{z})}\Big[\mathbb{E}_{p_d(\mathbf{x})}[E_\theta(\mathbf{x}|\mathbf{z})] \\
& - \mathbb{E}_{\mathrm{stop\_grad}(\mathrm{q}_\theta(\mathbf{x}))}[E_\theta(\mathbf{x},\mathbf{z})]\Big] + \mathbb{E}_{q_\theta(\mathbf{x})}\Big[\mathbb{E}_{q_{\mathrm{stop\_grad}(\phi)}(\mathbf{z})}\big[E_{\mathrm{stop\_grad}(\theta)}(\mathbf{x},\mathbf{z}) \\
& - \log p_{\mathrm{stop\_grad}(\phi)}(\mathbf{y}|\mathbf{z},\mathbf{x})\big] - \mathbb{D}_{\mathrm{KL}}[q_{\mathrm{stop\_grad}(\phi)}(\mathbf{z}|\mathbf{d_x})||p_{\mathrm{stop\_grad}(\phi)}(\mathbf{z}|\mathbf{x})]\Big],
\end{aligned}$$
$$(10)$$

where $p_d(\mathbf{x})$ and $q_\theta(\mathbf{x})$ denote the data distribution and the $t$ sequential MCMC samplings from the energy-based distribution of the source domain $D_s$. Similar to eq. (7), the first term trains the classification model on the source data. The second term trains the energy function to model the source data distribution. The last term is conducted on the adapted samples to supervise the adaptation procedure. The complete derivation is provided in Appendix A. An illustration of our model is shown in Figure 2. We also provide the complete algorithm in Appendix B.

**Ensemble inference.** Since the target data is inaccessible during training, we train the specific parameters $\theta$ and $\phi$ to model each source distribution by adapting the samples from other source domains to the current source distribution. In each iteration, we train the energy-based model $\theta^i$ of one randomly selected source domain $D_s^i$. The adapted samples generated by samples $\mathbf{x}^j, \mathbf{x}^k$ from the other source domains $D_s^j, D_s^k$ are used as the negative samples while $\mathbf{x}^i$ as the positive samples to train the energy-based model. During inference, the target sample is adapted to each source distribution with the specific energy function and predicted by the specific classifier. After that, we combine the predictions of all source domain models to obtain the final prediction:

$$p(\mathbf{y}_t) = \frac{1}{S}\sum_{i=1}^{S}\frac{1}{N}\sum_{n=1}^{N} p_{\phi^i}(\mathbf{y}|\mathbf{z}^n,\mathbf{x}) \qquad \mathbf{z}^n \sim p(\mathbf{z}^n|\mathbf{x}_t), \mathbf{x} \sim p_{\theta^i}(\mathbf{x}). \quad (11)$$

Here $\phi^i$ and $\theta^i$ denote the domain specific classification model and energy function of domain $D_s^i$. Note that since the labels of the $\mathbf{x}^t$ are unknown, $\mathbf{d}_{\mathbf{x}^t}$ in eq. (10) is not available during inference.

Table 1: **Benefit of energy-based test sample adaptation.** Experiments on PACS using a ResNet-18 averaged over five runs. Optimized by eq. (7), our model improves after adaptation. With the latent variable (eq. 10) performance improves further, both before and after adaptation.

| | Adaptation | **Photo** | **Art-painting** | **Cartoon** | **Sketch** | *Mean* |
|---|---|---|---|---|---|---|
| Without latent variable (eq. 7) | ✗ | 94.73 $\pm$0.22 | 78.66 $\pm$0.59 | 78.24 $\pm$0.71 | 78.34 $\pm$0.62 | 82.49 $\pm$0.26 |
| | ✓ | 94.59 $\pm$0.16 | 80.45 $\pm$0.52 | 79.98 $\pm$0.51 | **79.23** $\pm$0.32 | 83.51 $\pm$0.30 |
| With latent variable (eq. 10) | ✗ | 95.12 $\pm$0.41 | 79.79 $\pm$0.64 | 79.15 $\pm$0.37 | 79.28 $\pm$0.82 | 83.33 $\pm$0.43 |
| | ✓ | **96.05** $\pm$0.37 | **82.28** $\pm$0.31 | **81.55** $\pm$0.65 | **79.81** $\pm$0.41 | **84.92** $\pm$0.59 |

Therefore, we draw $\mathbf{z}^n$ from the prior distribution $p(\mathbf{z}^n|\mathbf{x}_t)$, where $\mathbf{x}_t$ is the original target sample without any adaptation. With fixed $\mathbf{z}^n$, $\mathbf{x} \sim p_{\theta^i}(\mathbf{x})$ are drawn by Langevin dynamics as in eq. (2) with the target samples $\mathbf{x}^t$ as the initialization sample. $p_{\theta^i}(\mathbf{x})$ denotes the distributions modeled by the energy function $E_{\theta^i}(\mathbf{x}|\mathbf{z}^n)$. Moreover, to be efficient, the feature extractor $\psi$ for obtaining feature representations $\mathbf{x}$ is shared by all source domains and only the energy functions and classifiers are domain specific. We deploy the energy-based model on feature representations for lighter neural networks of the domain-specific energy functions and classification models.

## 3 EXPERIMENTS

**Datasets.** We conduct our experiments on five widely used datasets for domain generalization, **PACS** (Li et al., 2017), **Office-Home** (Venkateswara et al., 2017), **DomainNet** (Peng et al., 2019), and **Rotated MNIST and Fashion-MNIST**. Since we conduct the energy-based distribution on the feature space, our method can also handle other data formats. Therefore, we also evaluate the method on **PHEME** (Zubiaga et al., 2016), a dataset for natural language processing.

PACS consists of 9,991 images of seven classes from four domains, i.e., *photo*, *art-painting*, *cartoon*, and *sketch*. We use the same training and validation split as (Li et al., 2017) and follow their "leave-one-out" protocol. Office-Home also contains four domains, i.e., *art*, *clipart*, *product*, and *real-world*, which totally have 15,500 images of 65 categories. DomainNet is more challenging since it has six domains i.e., *clipart*, *infograph*, *painting*, *quickdraw*, *real*, *sketch*, with 586,575 examples of 345 classes. We use the same experimental protocol as PACS. We utilize the Rotated MNIST and Fashion-MNIST datasets by following the settings in Piratla et al. (Piratla et al., 2020). The images are rotated from $0°$ to $90°$ in intervals of $15°$, covering seven domains. We use the domains with rotation angles from $15°$ to $75°$ as the source domains, and images rotated by $0°$ and $90°$ as the target domains. PHEME is a dataset for rumour detection. There are a total of 5,802 tweets labeled as rumourous or non-rumourous from 5 different events, i.e., *Charlie Hebdo*, *Ferguson*, *German Wings*, *Ottawa Shooting*, and *Sydney Siege*. Same as PACS, we evaluate our methods on PHEME also by the "leave-one-out" protocol.

**Implementation details.** We evaluate on PACS and Office-Home with both a ResNet-18 and ResNet-50 (He et al., 2016) and on DomainNet with a ResNet-50. The backbones are pretrained on ImageNet (Deng et al., 2009). On PHEME we conduct the experiments based on a pretrained DistilBERT (Sanh et al., 2019), following (Wright & Augenstein, 2020). To increase the number of sampling steps and sample diversity of the energy functions, we introduce a replay buffer $\mathcal{B}$ that stores the past updated samples from the modeled distribution (Du & Mordatch, 2019). The details of the models and hyperparameters are provided in Appendix C.

**Benefit of energy-based test sample adaptation.** We first investigate the effectiveness of our energy-based test sample adaptation in Table 1. Before adaptation, we evaluate the target samples directly by the classification model of each source domain and ensemble the predictions. After the adaptation to the source distributions, the performance of the target samples improves, especially on the *art-painting* and *cartoon* domains, demonstrating the benefit of the iterative adaptation by our energy-based model. The results of models with latent variable, i.e., trained by eq. (10), are shown in the last two rows. With the latent variable, the performance is improved both before and after adaptation, which shows the benefit of incorporating the latent variable into the classification model. The performance improvement after adaptation is also more prominent than without the latent variable, demonstrating the effectiveness of incorporating the latent variable into the energy function.

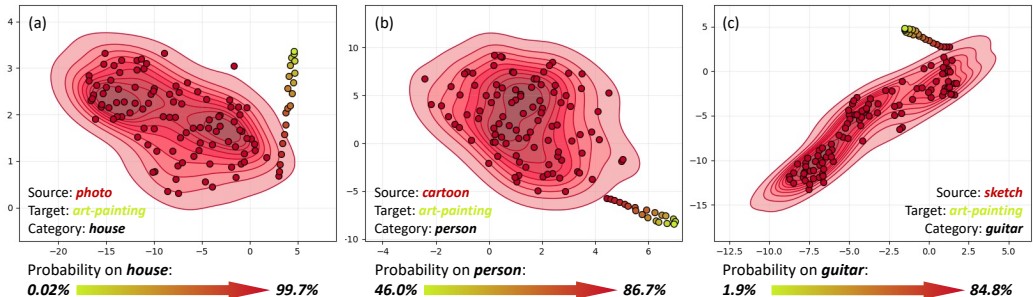

Figure 3: **Iterative adaptation of target samples.** We adapt the samples from the target domain (*art-painting*) to different source domains (*photo*, *cartoon*, and *sketch* in (a), (b), (c)). Each subfigure shows the adaptation of one target sample. The adaptation procedure of the target samples is represented by the gradient color from green to red. In each figure, the target sample has the same label as the source data, but is mispredicted due to domain shifts. During adaptation, the target samples gradually approach the source distributions and eventually get correct predictions.

**Effectiveness of iterative test sample adaptation by Langevin dynamics.** We visualize the iterative adaptation of the target samples in Figure 3. In each subfigure, the target and source samples have the same label. The visualization shows that the target samples gradually approach the source data distributions during the iterative adaptation by Langevin dynamics. After adaptation, the predictions of the target samples on the source domain classifier also become more accurate. For instance, in Figure 3 (a), the target sample of the *house* category is predicted incorrectly, with a probability of *house* being only 0.02%. After adaptation, the probability becomes 99.7%, which is predicted correctly. More visualizations, including several failure cases, are provided in Appendix D.

**Adaptation with different Langevin dynamics steps.** We also investigate the effect of the Langevin dynamics step numbers during adaptation. Figure 4 shows the variety of the average energy and accuracy of the target samples adapted to the source distributions with different updating steps. The experiments are conducted on PACS with ResNet-18. The target domain is *art-painting*. With the step numbers less than 80, the average energy decreases consistently while the accuracy increases along with the increased number of updating steps, showing that the target samples are getting closer to the source distributions. When the step numbers are too large, the accuracy will decrease as the number of steps increases. We attribute this to $\mathbf{z}_t$ having imperfect categorical information, since it is approximated during inference from a single target sample $\mathbf{x}_t$ only. In this case, the label information would not be well preserved in $\mathbf{x}_t$

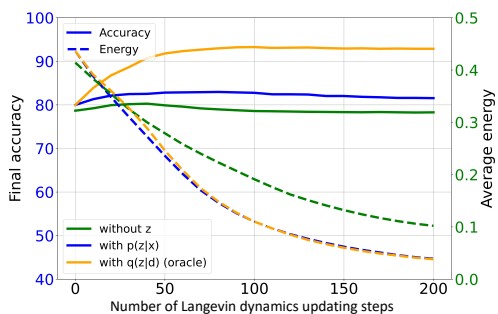

Figure 4: **Adaptation with different Langevin dynamics steps.** As the number of steps increases, energy decreases while accuracy increases. When the number of steps is too large, the accuracy without $\mathbf{z}$ or with $p(\mathbf{z}|\mathbf{x})$ drops slightly while the accuracy with $q(\mathbf{z}|\mathbf{d}_\mathbf{x})$ is more stable and better.

during the Langevin dynamics update, which causes an accuracy drop with a large number of updates. To demonstrate this, we conduct the experiment by replacing $p(\mathbf{z}_t)$ with $q_\phi(\mathbf{z}|\mathbf{d}_\mathbf{x})$ during inference. $\mathbf{d}_\mathbf{x}$ is the class center of the same class as $\mathbf{x}_t$. Therefore, $q_\phi(\mathbf{z}|\mathbf{d}_\mathbf{x})$ contains categorical information that is closer to the ground truth label. We regard this as the oracle model. As expected, the oracle model performs better as the number of steps increases and reaches stability after 100 steps. We also show the results without $\mathbf{z}$. We can see the performance and stability are both worse, which again demonstrates that $\mathbf{z}$ helps preserve label information in the target samples during adaptation. Moreover, the energy without conditioning on $\mathbf{z}$ is higher. The reason can be that without conditioning on $\mathbf{z}$, there is no guidance of categorical information during sample adaptation. In this case, the sample can be adapted randomly by the energy-based model, regardless of the categorical information. This can lead to the conflict to adapt the target features to different categories of the source data, slowing

Table 2: **Comparisons on image and text datasets.** Our method achieves the best mean accuracy for all datasets, independent of the backbone. Larger adaptation steps (i.e., 50) lead to better performance.

| | PACS | | Office-Home | | DomainNet | PHEME |
|---|---|---|---|---|---|---|
| | ResNet-18 | ResNet-50 | ResNet-18 | ResNet-50 | ResNet-50 | DistilBERT |
| Iwasawa & Matsuo (2021) | 81.40 | 85.10 | 57.00 | 68.30 | - | - |
| Zhou et al. (2020a) | 82.83 | 84.90 | 65.63 | 67.66 | - | - |
| Gulrajani & Lopez-Paz (2020) | - | 85.50 | - | 66.50 | 40.90 | - |
| Wang et al. (2021) | 83.09 | 86.23 | 64.13 | 67.99 | - | 75.8 $\pm 0.23$ |
| Dubey et al. (2021) | - | | - | 68.90 | 43.90 | - |
| Xiao et al. (2022) | 84.15 | 87.51 | 66.02 | 71.07 | - | 76.1 $\pm 0.21$ |
| *This paper w/o adaptation* | 83.33 $\pm 0.43$ | 86.05 $\pm 0.37$ | 65.01 $\pm 0.47$ | 70.44 $\pm 0.25$ | 42.90 $\pm 0.34$ | 75.4 $\pm 0.13$ |
| *This paper w/ adaptation (10 steps)* | 84.25 $\pm 0.48$ | 87.05 $\pm 0.26$ | 65.73 $\pm 0.32$ | 71.13 $\pm 0.43$ | 43.75 $\pm 0.49$ | 76.0 $\pm 0.16$ |
| *This paper w/ adaptation (20 steps)* | **84.92** $\pm 0.59$ | 87.70 $\pm 0.28$ | 66.31 $\pm 0.21$ | **72.07** $\pm 0.38$ | 44.66 $\pm 0.51$ | 76.5 $\pm 0.18$ |
| *This paper w/ adaptation (50 steps)* | **85.10** $\pm 0.33$ | **88.12** $\pm 0.25$ | **66.75** $\pm 0.21$ | **72.25** $\pm 0.32$ | **44.98** $\pm 0.43$ | **76.9** $\pm 0.16$ |

down the decline of the energy. We provide more analyses of $\mathbf{z}_t$ in Appendix E. In addition, the training and test time cost is also larger as the step number increases, the comparisons and analyses are also provided in Appendix E.

**Comparisons.** PACS, Office-Home, and DomainNet are three widely used benchmarks in domain generalization. We conduct experiments on PACS and Office-Home based on both ResNet-18 and ResNet-50 and experiments on DomainNet based on ResNet-50. As shown in Table 2, our method achieves competitive and even the best overall performance in most cases. Moreover, our method performs better than most of the recent test-time adaptation methods (Iwasawa & Matsuo, 2021; Wang et al., 2021; Dubey et al., 2021), which fine-tunes the model at test time with batches of target samples. By contrast, we strictly follow the setting of domain generalization. We only use the source data to train the classification and energy-based models during training. At test time, we do our sample adaptation and make predictions on each individual target sample by just the source-trained models. Our method is more data efficient at test time, avoiding the problem of data collection per target domain in real-world applications. Despite the data efficiency during inference, our method is still comparable and sometimes better, especially on datasets with more categories, e.g., Office-Home and DomainNet. Compared with the recent work by Xiao et al. (2022), our method is at least competitive and often better. To show the generality of our method, we also conduct experiments on the natural language processing dataset PHEME. The dataset is a binary classification task for rumour detection. The results in Table 2 show similar conclusions as the image datasets.

Table 2 also demonstrates the effectiveness of our sample adaptation. For each dataset and backbone, the proposed method achieves a good improvement after adaptation by the proposed discriminative energy-based model. For fairness, the results without adaptation are also obtained by ensemble predictions of the source-domain-specific classifiers. Moreover, larger steps (i.e., 50) lead to better performance. The improvements of adaptation with 50 steps are slight. Considering the trade-off of computational efficiency and performance, we set the step number as 20 in our paper. We provide detailed comparisons, results on rotated MNIST and Fashion-MNIST datasets, as well as more experiments on the latent variable, corruption datasets, and analyses of the ensemble inference method in Appendix E.

## 4 RELATED WORK

**Domain generalization.** One of the predominant methods is domain invariant learning (Muandet et al., 2013; Ghifary et al., 2016; Motiian et al., 2017; Seo et al., 2020; Zhao et al., 2020; Xiao et al., 2021a; Mahajan et al., 2021; Nguyen et al., 2021; Phung et al., 2021; Shi et al., 2022). Muandet et al. (2013) and Ghifary et al. (2016) learn domain invariant representations by matching the moments of features across source domains. Li et al. (Li et al., 2018b) further improved the model by learning conditional-invariant features. Recently, Mahajan et al. (2021) introduced causal matching to model within-class variations for generalization. Shi et al. (2022) provided a gradient matching to encourage consistent gradient directions across domains. Arjovsky et al. (2019) and Ahuja et al. (2021) proposed invariant risk minimization to learn an invariant classifier. Another widely used methodology is domain augmentation (Shankar et al., 2018; Volpi et al., 2018; Qiao et al., 2020; Zhou et al., 2020a;b; Yao et al., 2022), which generates more source domain data to simulate domain shifts during training.

Zhou et al. (2020b) proposed a data augmentation on the feature space by mixing the feature statistics of instances from different domains. Meta-learning-based methods have also been studied for domain generalization (Li et al., 2018a; Balaji et al., 2018; Dou et al., 2019; Du et al., 2021b; Bui et al., 2021; Du et al., 2021c). Li et al. (2018a) introduced the model agnostic meta-learning (Finn et al., 2017) into domain generalization. Du et al. (2020b) proposed the meta-variational information bottleneck for domain-invariant learning.

**Test-time adaptation and source-free adaptation.** Recently, adaptive methods have been proposed to better match the source-trained model and the target data at test time (Sun et al., 2020; Li et al., 2020; D'Innocente et al., 2019; Pandey et al., 2021; Iwasawa & Matsuo, 2021; Dubey et al., 2021; Zhang et al., 2021). Test-time adaptation (Sun et al., 2020; Wang et al., 2021; Liu et al., 2021; Zhou & Levine, 2021) fine-tunes (part of) a network trained on source domains by batches of target samples. Xiao et al. (2022) proposed single-sample generalization that adapts a model to each target sample under a meta-learning framework. There are also some source-free domain adaptation methods (Liang et al., 2020; Yang et al., 2021; Dong et al., 2021; Liang et al., 2021) that adapt the source-trained model on only the target data. These methods follow the domain adaptation settings to fine-tune the source-trained model by the entire target set. By contrast, we do sample adaptation at test time but strictly follow the domain generalization settings. In our method, no target sample is available during the training of the models. At test time, each target sample is adapted to the source domains and predicted by the source-trained model individually, without fine-tuning the models.

**Energy-based model.** The energy-based model is a classical learning framework (Ackley et al., 1985; Hinton, 2002; Hinton et al., 2006; LeCun et al., 2006). Recently, (Xie et al., 2016; Nijkamp et al., 2019; 2020; Du & Mordatch, 2019; Du et al., 2021a; Xie et al., 2022) further extend the energy-based model to high-dimensional data using contrastive divergence and Stochastic Gradient Langevin dynamics. Wang et al. (2023) utilize the energy-based model for effective self-supervised pretraining of vision models. Different from most of these works that model the data distributions, some recent works model the joint distributions (Grathwohl et al., 2020; Xiao et al., 2021b). In our work, we define the joint distribution of data and label to promote the classification of unseen target samples in domain generalization. We further incorporate a latent variable to incorporate the categorical information into the Langevin dynamics procedure. Energy-based models for various tasks have been proposed, e.g., image generation (Du et al., 2020a; Nie et al., 2021), out-of-distribution detection (Liu et al., 2020), and anomaly detection (Dehaene et al., 2020; Wang et al., 2022). Some methods also utilize energy-based models for domain adaptation (Zou et al., 2021; Xie et al., 2021; Kurmi et al., 2021). Different from these methods, we focus on domain generalization and utilize the energy-based model to express the source domain distributions without any target data during training.

## 5 CONCLUSION AND DISCUSSIONS

In this paper, we propose a discriminative energy-based model to adapt the target samples to the source data distributions for domain generalization. The energy-based model is designed on the joint space of input, output, and a latent variable, which is constructed by a domain specific classification model and an energy function. With the trained energy-based model, the target samples are adapted to the source distributions through Langevin dynamics and then predicted by the classification model. Since we aim to prompt the classification of the target samples, the model is trained to achieve label-preserving adaptation by incorporating the categorical latent variable. We evaluate the method on six image and text benchmarks. The results demonstrate its effectiveness and generality. We have not tested our approach beyond image and text classification tasks, but since our sample adaptation is conducted on the feature space, it should be possible to extend the method to other complex tasks based on feature representations.

Compared with recent model adaptation methods, our method does not need to adjust the model parameters at test time, which requires batches of target samples to provide sufficient target information. This is more data efficient and challenging at test time, therefore the training procedure is more involved with complex optimization objectives. One limitation of our proposed method is the iterative adaptation requirement for each target sample, which introduces an extra time cost at both training and test time. The problem can be mitigated by speeding up the energy minimization with optimization techniques during Langevin dynamics, e.g., Nesterov momentum (Nesterov, 1983), or by exploring one-step methods for sample adaptation. We leave these explorations for future work.

ACKNOWLEDGMENT

This work is financially supported by the Inception Institute of Artificial Intelligence, the University of Amsterdam and the allowance Top consortia for Knowledge and Innovation (TKIs) from the Netherlands Ministry of Economic Affairs and Climate Policy.

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

## A  DERIVATIONS

**Derivation of energy-based sample adaptation.** Recall our discriminative energy-based model

$$p_{\theta,\phi}(\mathbf{x}, \mathbf{y}) = p_\phi(\mathbf{y}|\mathbf{x})\frac{\exp(-E_\theta(\mathbf{x}))}{Z_{\theta,\phi}}, \tag{12}$$

where $Z_{\theta,\phi} = \int p_\phi(\mathbf{y}|\mathbf{x})\exp(-E_\theta(\mathbf{x}))d\mathbf{x}d\mathbf{y}$ is the partition function. $\phi$ and $\theta$ denote the parameters of the classifier and energy function, respectively. To jointly train the parameters, we minimize the contrastive divergence proposed by Hinton (2002):

$$\mathcal{L} = \mathbb{D}_{\text{KL}}[p_d(\mathbf{x}, \mathbf{y})||p_{\theta,\phi}(\mathbf{x}, \mathbf{y})] - \mathbb{D}_{\text{KL}}[q_{\theta,\phi}(\mathbf{x}, \mathbf{y})||p_{\theta,\phi}(\mathbf{x}, \mathbf{y})], \tag{13}$$

where $p_d(\mathbf{x}, \mathbf{y})$ denotes the real data distribution and $q_{\theta,\phi}(\mathbf{x}, \mathbf{y}) = \prod_\theta^t p(\mathbf{x}, \mathbf{y})$ denotes $t$ sequential MCMC samplings from the distribution expressed by the energy-based model (Du et al., 2021a). The gradient of the first term with respect to $\theta$ and $\phi$ is

$$\begin{aligned}
\nabla_{\theta,\phi}\mathbb{D}_{\text{KL}}[p_d(\mathbf{x}, \mathbf{y})||p_{\theta,\phi}(\mathbf{x}, \mathbf{y})] &= \nabla_{\theta,\phi}\mathbb{E}_{p_d(\mathbf{x},\mathbf{y})}\big[\log\frac{p_d(\mathbf{x}, \mathbf{y})}{p_{\theta,\phi}(\mathbf{x}, \mathbf{y})}\big] \\
&= \mathbb{E}_{p_d(\mathbf{x},\mathbf{y})}\big[\nabla_{\theta,\phi}\log p_d(\mathbf{x}, \mathbf{y}) - \nabla_{\theta,\phi}\log p_{\theta,\phi}(\mathbf{x}, \mathbf{y})\big] \\
&= \mathbb{E}_{p_d(\mathbf{x},\mathbf{y})}\big[-\nabla_{\theta,\phi}\log p_{\theta,\phi}(\mathbf{x}, \mathbf{y})\big],
\end{aligned} \tag{14}$$

while the gradient of the second term is

$$\begin{aligned}
&\nabla_{\theta,\phi}\mathbb{D}_{\text{KL}}[q_{\theta,\phi}(\mathbf{x}, \mathbf{y})||p_{\theta,\phi}(\mathbf{x}, \mathbf{y})] \\
=&\nabla_{\theta,\phi}\mathbb{E}_{q_{\theta,\phi}(\mathbf{x},\mathbf{y})}\big[\log\frac{q_{\theta,\phi}(\mathbf{x}, \mathbf{y})}{p_{\theta,\phi}(\mathbf{x}, \mathbf{y})}\big] \\
=&\nabla_{\theta,\phi}q_{\theta,\phi}(\mathbf{x}, \mathbf{y})\nabla_{q_{\theta,\phi}}\mathbb{D}_{\text{KL}}[q_{\theta,\phi}(\mathbf{x}, \mathbf{y})||p_{\theta,\phi}(\mathbf{x}, \mathbf{y})] + \mathbb{E}_{q_{\theta,\phi}(\mathbf{x},\mathbf{y})}\big[-\nabla_{\theta,\phi}\log p_{\theta,\phi}(\mathbf{x}, \mathbf{y})\big].
\end{aligned} \tag{15}$$

Combining eq. (14) and eq. (15), we have the overall gradient as:

$$\begin{aligned}
\nabla_{\theta,\phi}\mathcal{L}_{all} = -\big(&\mathbb{E}_{p_d(\mathbf{x},\mathbf{y})}\big[\nabla_{\theta,\phi}\log p_{\theta,\phi}(\mathbf{x}, \mathbf{y})\big] - \mathbb{E}_{q_{\theta,\phi}(\mathbf{x},\mathbf{y})}\big[\nabla_{\theta,\phi}\log p_{\theta,\phi}(\mathbf{x}, \mathbf{y})\big] \\
&+ \nabla_{\theta,\phi}q_{\theta,\phi}(\mathbf{x}, \mathbf{y})\nabla_{q_{\theta,\phi}}\mathbb{D}_{\text{KL}}[q_{\theta,\phi}(\mathbf{x}, \mathbf{y})||p_{\theta,\phi}(\mathbf{x}, \mathbf{y})]\big).
\end{aligned} \tag{16}$$

For the first two terms, the gradient can be further derived to

$$\begin{aligned}
&\mathbb{E}_{p_d(\mathbf{x},\mathbf{y})}\big[\nabla_{\theta,\phi}\log p_{\theta,\phi}(\mathbf{x}, \mathbf{y})\big] - \mathbb{E}_{q_{\theta,\phi}(\mathbf{x},\mathbf{y})}\big[\nabla_{\theta,\phi}\log p_{\theta,\phi}(\mathbf{x}, \mathbf{y})\big] \\
=&\mathbb{E}_{p_d(\mathbf{x},\mathbf{y})}\big[\nabla_{\theta,\phi}(\log p_\phi(\mathbf{y}|\mathbf{x}) - E_\theta(\mathbf{x}) - \log Z_{\theta,\phi})\big] \\
&-\mathbb{E}_{q_{\theta,\phi}(\mathbf{x},\mathbf{y})}\big[\nabla_{\theta,\phi}(\log p_\phi(\mathbf{y}|\mathbf{x}) - E_\theta(\mathbf{x}) - \log Z_{\theta,\phi})\big].
\end{aligned} \tag{17}$$

Moreover, $\nabla_{\theta,\phi}\log Z_{\theta,\phi}$ can be written as the expectation $\mathbb{E}_{p_{\theta,\phi}(\mathbf{x},\mathbf{y})}[\nabla_\phi\log p_\phi(\mathbf{y}|\mathbf{x}) - \nabla_\theta E_\theta(\mathbf{x})]$ (Song & Kingma, 2021; Xiao et al., 2021b), which is therefore canceled out in eq. (17) (Hinton, 2002). We then have the loss function for the first two terms as

$$\mathcal{L}_1 = \mathbb{E}_{p_d(\mathbf{x},\mathbf{y})}\big[E_\theta(\mathbf{x}) - \log p_\phi(\mathbf{y}|\mathbf{x})\big] - \mathbb{E}_{q_{\theta,\phi}(\mathbf{x},\mathbf{y})}\big[E_\theta(\mathbf{x}) - \log p_\phi(\mathbf{y}|\mathbf{x})\big]. \tag{18}$$

Furthermore, we have the loss function

$$\begin{aligned}
\mathcal{L}_2 &= \mathbb{E}_{q_{\theta,\phi}(\mathbf{x},\mathbf{y})}\big[\log\frac{q_{\theta,\phi}(\mathbf{x}, \mathbf{y})}{p_{stop\_grad(\theta,\phi)}(\mathbf{x}, \mathbf{y})}\big] \\
&= -\mathbb{E}_{q_{\theta,\phi}(\mathbf{x},\mathbf{y})}\big[\log p_{stop\_grad(\phi)}(\mathbf{y}|\mathbf{x}) - E_{stop\_grad(\theta)}(\mathbf{x}) - \log Z_{stop\_grad(\theta,\phi)}\big] \\
&\quad + \mathbb{E}_{q_{\theta,\phi}(\mathbf{x},\mathbf{y})}\big[\log q_{\theta,\phi}(\mathbf{x}, \mathbf{y})\big],
\end{aligned} \tag{19}$$

which has the same gradient as the last term in eq. (16) (Du et al., 2021a). The $stop\_grad$ here means that we do not backpropagate the gradients to update the parameters by the corresponding forward functions. Thus, these parameters can be treated as constants.

Since the gradient of $\theta$ and $\phi$ is stopped in $\log Z_{stop\_grad(\theta,\phi)}$, we treat it as a constant independent of $q_{\theta,\phi}(\mathbf{x}, \mathbf{y})$ and therefore remove it from the eq. (19). In addition, the term $\mathbb{E}_{q_{\theta,\phi}(\mathbf{x},\mathbf{y})}\big[\log p_\phi(\mathbf{y}|\mathbf{x})\big]$ in eq. (18) encourages wrong prediction of the updated samples from $q_{\theta,\phi}(\mathbf{x}, \mathbf{y})$, which goes against

our goal of promoting classification by adapting target samples. The term $\mathbb{E}_{q_{\theta,\phi}(\mathbf{x},\mathbf{y})}\big[\log q_{\theta,\phi}(\mathbf{x},\mathbf{y})\big]$ in eq. (19) can be treated as a negative entropy of $q_{\theta,\phi}(\mathbf{x},\mathbf{y})$, which is always negative and hard to estimate. Therefore, we remove these two terms in the final loss function by applying an upper bound of the combination of eq. (18) and eq. (19) as:

$$\mathcal{L} = -\mathbb{E}_{p_d(\mathbf{x},\mathbf{y})}\big[\log p_\phi(\mathbf{y}|\mathbf{x})\big] + \mathbb{E}_{p_d(\mathbf{x},\mathbf{y})}\big[E_\theta(\mathbf{x})\big] - \mathbb{E}_{\text{stop\_grad}(q_{\theta,\phi}(\mathbf{x},\mathbf{y}))}\big[E_\theta(\mathbf{x})\big]$$
$$+ \mathbb{E}_{q_{\theta,\phi}(\mathbf{x},\mathbf{y})}\big[E_{\text{stop\_grad}(\theta)}(\mathbf{x}) - \log p_{\text{stop\_grad}(\phi)}(\mathbf{y}|\mathbf{x})\big]. \tag{20}$$

**Energy-based sample adaptation with categorical latent variable.** To keep the categorical information during sample adaptation, we introduce a categorical latent variable $\mathbf{z}$ into our discriminative energy-based model, which is defined as $p_{\theta,\phi}(\mathbf{x},\mathbf{y}) = \int p_{\theta,\phi}(\mathbf{x},\mathbf{y},\mathbf{z})d\mathbf{z} = \int p_\phi(\mathbf{y}|\mathbf{z},\mathbf{x})p_\phi(\mathbf{z}|\mathbf{x})\frac{\exp(-E_\theta(\mathbf{x}|\mathbf{z}))}{Z_{\theta,\phi}}d\mathbf{z}$. We optimize the parameters $\theta$ and $\phi$ also by the contrastive divergence $\mathbb{D}_{\text{KL}}[p_d(\mathbf{x},\mathbf{y})||p_{\theta,\phi}(\mathbf{x},\mathbf{y})] - \mathbb{D}_{\text{KL}}[q_{\theta,\phi}(\mathbf{x},\mathbf{y})||p_{\theta,\phi}(\mathbf{x},\mathbf{y})]$, which has similar gradient as eq. (14) and eq. (15). The latent variable $\mathbf{z}$ is estimated by variational inference, leading to a lower bound of $\log p_{\theta,\phi}(\mathbf{x},\mathbf{y}) \geq \mathbb{E}_{q_\phi(\mathbf{z})}[\log p_\phi(\mathbf{y}|\mathbf{z},\mathbf{x}) - E_\theta(\mathbf{x}|\mathbf{z}) - \log Z_{\theta,\phi}] + \mathbb{D}_{\text{KL}}[q_\phi(\mathbf{z}|\mathbf{d_x})||p_\phi(\mathbf{z}|\mathbf{x})]$. We obtain the final loss function of the contrastive divergence in a similar way as eq. (20) by estimating the gradient and remove the terms that are hard to estimate or conflict with our final goal. The final objective function is:

$$\mathcal{L}_f = \mathbb{E}_{p_d(\mathbf{x},\mathbf{y})}\Big[\mathbb{E}_{q_\phi(\mathbf{z})}[-\log p_\phi(\mathbf{y}|\mathbf{z},\mathbf{x})] + \mathbb{D}_{\text{KL}}[q_\phi(\mathbf{z}|\mathbf{d_x})||p_\phi(\mathbf{z}|\mathbf{x})]\Big] + \mathbb{E}_{q_\phi(\mathbf{z})}\Big[\mathbb{E}_{p_d(\mathbf{x})}[E_\theta(\mathbf{x}|\mathbf{z})]$$

$$- \mathbb{E}_{\text{stop\_grad}(q_\theta(\mathbf{x}))}[E_\theta(\mathbf{x},\mathbf{z})]\Big] + \mathbb{E}_{q_\theta(\mathbf{x})}\Big[\mathbb{E}_{q_{\text{stop\_grad}(\phi)}(\mathbf{z})}\big[E_{\text{stop\_grad}(\theta)}(\mathbf{x},\mathbf{z})$$

$$- \log p_{\text{stop\_grad}(\phi)}(\mathbf{y}|\mathbf{z},\mathbf{x})\big] - \mathbb{D}_{\text{KL}}[q_{\text{stop\_grad}(\phi)}(\mathbf{z}|\mathbf{d_x})||p_{\text{stop\_grad}(\phi)}(\mathbf{z}|\mathbf{x})]\Big]. \tag{21}$$

## B  ALGORITHM

We provide the detailed training and test algorithm of our energy-based sample adaptation in Algorithm 1.

## C  DATASETS AND IMPLEMENTATION DETAILS

**Model.** To be efficient, we train a shared backbone for all source domains while a domain-specific classifier and a neural-network-based energy function for each source domain. The feature extractor backbone is a basic Residual Network without the final fully connected layer (classifier). Both the prior distribution $p_\phi(\mathbf{z}|\mathbf{x})$ and posterior distribution $q_\phi(\mathbf{z}|\mathbf{d_x})$ of the latent variable $\mathbf{z}$ are generated by a neural network $\phi$ that consists of four fully connected layers with ReLU activation, which outputs the mean and variance of the distribution. The last layer of $\phi$ outputs both the mean and standard derivation of the distribution $p_\phi(\mathbf{z}|\mathbf{x})$ and $q_\phi(\mathbf{z}|\mathbf{d_x})$ for further Monte Carlo sampling. The dimension of $\mathbf{z}$ is the same as the feature representations $\mathbf{x}$, e.g., 512 for ResNet-18 and 2048 for ResNet-50. $\mathbf{d_x}$ is obtained by the center features of the batch of samples that have the same categories as the current sample $\mathbf{x}$ in each iteration.

Deployed on feature representations, the energy function consists of three fully connected layers with two dropout layers. The latent variable $\mathbf{z}$ is incorporated into the energy function by concatenating with the feature representation $\mathbf{x}$. The input dimension is doubled of the output feature of the backbone, i.e., 1024 for ResNet-18 and 4096 for ResNet-50. We use the *swish* function as activation in the energy functions (Du et al., 2021a). The final output of the EBM is a scalar, which is processed by a sigmoid function following Du et al. (2021a) to bound the energy to the region $[0, 1]$ and improve the stability during training. During training, we introduce a replay buffer $\mathcal{B}$ to store the past updated samples from the modeled distribution (Du & Mordatch, 2019). By sampling from $\mathcal{B}$ with 50% probability, we can initialize the negative samples with either the sample features from other source domains or the past Langevin dynamics procedure. This can increase the number of sampling steps and the sample diversity.

**Training details and hyperparameters.** We evaluate on PACS with both a ResNet-18 and ResNet-50 pretrained on ImageNet. We use Adam optimization and train for 10,000 iterations with a batch size

---

**Algorithm 1** Energy-based sample adaptation

TRAINING TIME

**Require:** Source domains $\mathcal{D}_s = \left\{ D_s^i \right\}_{i=1}^S$ each with joint distribution $p_{d_s^i}(I, \mathbf{y})$ of input image and label.

**Require:** Learning rate $\mu$; iteration numbers $M$; step numbers $K$ and step size $\lambda$ of the energy function.

Initialize pretrained backbone $\psi$; $\phi^i, \theta^i, \mathcal{B}^i = \varnothing$ for each source domain $D_s^i$.

**for** *iter* in $M$ **do**
    **for** $D_s^i$ in $\mathcal{D}_s$ **do**
        Sample datapoints $\{(I^i, \mathbf{y}^i)\} \sim p_{d_s^i}(I, \mathbf{y})$; $\{(I^j, \mathbf{y}^j)\} \sim \{p_{d_s^j}(I, \mathbf{y})\}_{j \neq i}$ or $\mathcal{B}$ with 50% probability.
        Feature representations $\mathbf{x}^i = f_\psi(I^i), \mathbf{x}^j = f_\psi(I^j)$
        **for** $k$ in $K$ **do**
            $\mathbf{x}_k^j \leftarrow \mathbf{x}_{k-1}^j - \nabla_\mathbf{x} E_{\theta^i}(\mathbf{x}_{k-1}^j | \mathbf{z}^j) + \omega, \quad \mathbf{z}^j \sim q_{\phi^i}(\mathbf{z}^j | \mathbf{d}_{\mathbf{x}^j}), \quad \omega \sim \mathcal{N}(0, \sigma).$
        **end for**
        $q_{\theta^i}(\mathbf{x}) \leftarrow \mathbf{x}_k^j, \quad p_d(\mathbf{x}) \leftarrow p_{d_s^i}(\mathbf{x}^i).$
        $(\psi, \phi^i) \leftarrow (\psi, \phi^i) - \lambda \nabla_{\psi, \phi^i} \mathcal{L}_f(p_d(\mathbf{x})) \quad \theta^i \leftarrow \theta^i - \lambda \nabla_{\theta^i} \mathcal{L}_f(p_d(\mathbf{x}), q_{\theta^i}(\mathbf{x})).$
        $\mathcal{B}^i \leftarrow \mathcal{B}^i \cup \mathbf{x}_k^j$
    **end for**
**end for**

---

TEST TIME

**Require**: Target images $I_t$ from the target domain; trained backbone $\psi$; and domain-specific model $\phi^i, \theta^i$ for each source domain in $\left\{ D_s^i \right\}_{i=1}^S$.

Input feature representations $\mathbf{x}_t = f_\psi(I_t)$.

**for** $i$ in $\{1, \ldots, S\}$ **do**
    **for** $k$ in $K$ **do**
        $\mathbf{x}_{t,k} \leftarrow \mathbf{x}_{t,k-1} - \nabla_\mathbf{x} E_{\theta^i}(\mathbf{x}_{t,k-1} | \mathbf{z}_t) + \omega, \quad \mathbf{z}_t \sim p_{\phi^i}(\mathbf{z}_t | \mathbf{x}_t), \quad \omega \sim \mathcal{N}(0, \sigma).$
    **end for**
    $\mathbf{y}_t^i = p_{\phi^i}(\mathbf{y}_t | \mathbf{x}_{t,k}, \mathbf{z}_t)$
**end for**
**return** $\mathbf{y}_t = \frac{1}{S} \sum_{i=1}^S \mathbf{y}_t^i.$

---

Table 3: **Implementation details** of our method per dataset and backbone.

| Dataset | Backbone | Backbone learning rate | Step size | Number of steps |
|---|---|---|---|---|
| PACS | ResNet-18 | 0.00005 | 50 | 20 |
| | ResNet-50 | 0.00001 | 50 | 20 |
| Office-Home | ResNet-18 | 0.00001 | 100 | 20 |
| | ResNet-50 | 0.00001 | 100 | 20 |
| Rotated MNIST | ResNet-18 | 0.00005 | 50 | 20 |
| Fashion-MNIST | ResNet-18 | 0.00005 | 50 | 20 |
| PHEME | DistilBERT | 0.00003 | 40 | 20 |

of 128. We set the learning rate to 0.00005 for ResNet-18, 0.00001 for ResNet-50, and 0.0001 for the energy-based model and classification model. We use 20 steps of Langevin dynamics sampling to adapt the target samples to source distributions, with a step size of 50. We set the number of Monte Carlo sampling $N$ in eq. (11) as 10 for PACS. Most of the experimental settings on Office-Home are the same as on PACS. The learning rate of the backbone is set to 0.00001 for both ResNet-18 and ResNet-50. The number of Monte Carlo sampling is 5. For fair comparison, we evaluate the rotated MNIST and Fashion-MNIST with ResNet-18, following (Piratla et al., 2020). The other settings are also the same as PACS. On PHEME we conduct the experiments based on a pretrained DistilBERT. We set the learning rate as 0.00003 and use 20 steps of Langevin dynamics with a step size of 20.

We train all models on an NVIDIA Tesla V100 GPU for 10,000 iterations. The learning rates of the backbone are different for different datasets as shown in Table 3. The learning rates of the domain-specific classifiers and energy functions are both set to 0.0001 for all datasets. For each source domain, we randomly select 128 samples as a batch to train the backbone and classification model. We also select 128 samples from the other source domains together with the current domain samples to train the domain-specific energy function. We use a replay buffer with 500 feature representations and apply spectral normalization on all weights of the energy function (Du & Mordatch, 2019). We

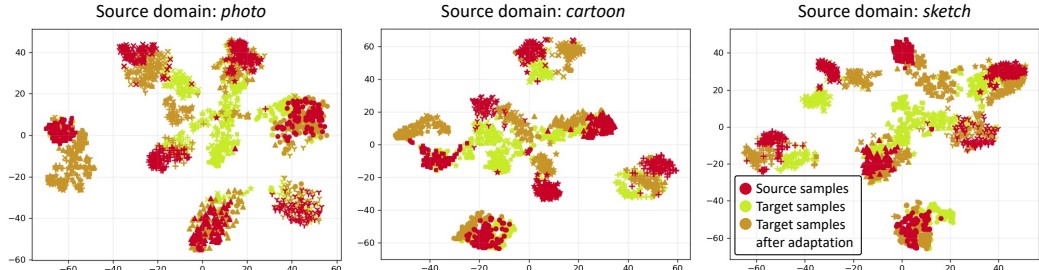

Figure 5: **Benefit of energy-based test sample adaptation.** Different shapes denote different classes. From left to right: adaptation to the source domains *photo*, *cartoon*, and *sketch* of samples from the target domain *art-painting*. After adaptation, the target samples (●) are more close to the source data (●) than before (●), demonstrating the effectiveness of our method. Best viewed in color.

use random noise with standard deviation $\lambda = 0.001$ and clip the gradients to have individual value magnitude of less than 0.01 similar to (Du & Mordatch, 2019). The step size and number of steps for Langevin dynamics are different for different datasets as shown in Table 3.

## D VISUALIZATIONS

**More visualizations of the adaptation procedure.** To further show the effectiveness of the iterative adaptation of target samples, we provide more visualizations on PACS. Figure 5 visualizes the source domain features and the target domain features both before and after the adaptation to each individual source domain. Figure 6 visualizes more iterative adaptation procedure of the target samples. Subfigures in different rows show the adaptation of samples from different target domains to source domains. Similar with the visualizations in the main paper, the target samples gradually approach the source data distributions during the iterative adaptation. Therefore, the predictions of the target samples on the source domain classifier become more accurate after adaptation.

**Failure cases.** We also provide some failure cases on PACS in Figure 7 to gain more insights in our method. Our method is confused with samples that have objects of different categories (first row) and multiple objects or complex background (last three rows). A possible reason is that there is noisy information contained in the latent variable of these samples, leading to adaptation without a clear direction, which behaves as wrong adaptation directions, e.g., visualization in row 1 column 4, or unstable updates with fluctuations in small regions, e.g., visualizations in row 2 column 3 and row 3 column 4. Obtaining the latent variable with more accurate and clear categorical information can be one solution for these failure cases. We can also solve the problem by achieving more stable adaptations with optimization techniques like Nesterov momentum (Nesterov, 1983). Moreover, although failing in these cases, the adaptation of the target sample to some source domains still improves the performance, e.g., the adaptation of the *photo* sample (row 1 column 2) and *cartoon* sample (row 3 column 3) to the *art-painting* domain and the adaptation of the *sketch* sample (row 4 column 4) to the *cartoon* domain, which further demonstrate the effectiveness of our iterative sample adaptation through the energy-based model. The results motivate another solution for these failure cases, which is to learn to select the best source domain, or top-n source domains for adaptation and prediction of each target sample. We leave these explorations for future work.

## E MORE EXPERIMENTAL RESULTS

**Analyses and discussions of the categorical latent variables.** In the proposed method, the categorical latent representation for the test sample will have high fidelity to the correct class. This is guaranteed by the training procedure of our method. As shown in the training objective function (eq. 10), we minimize the KL divergence to encourage the prior $p_\phi(\mathbf{z}|\mathbf{x})$ to be close to the variational posterior $q_\phi(\mathbf{z}|\mathbf{d_x})$. $\mathbf{d_x}$ is essentially the class prototype containing the categorical information. By doing so, we train the inference model $p_\phi(\mathbf{z}|\mathbf{x})$ to learn to extract categorical information from a single sample. Moreover, we also supervise the sample adaptation procedures by the predicted log-likelihood of the adapted samples (the last term in eq. (10)). The supervision is inherent in the objective function of our discriminative energy-based model as in the derivation of eq. (7) and eq.

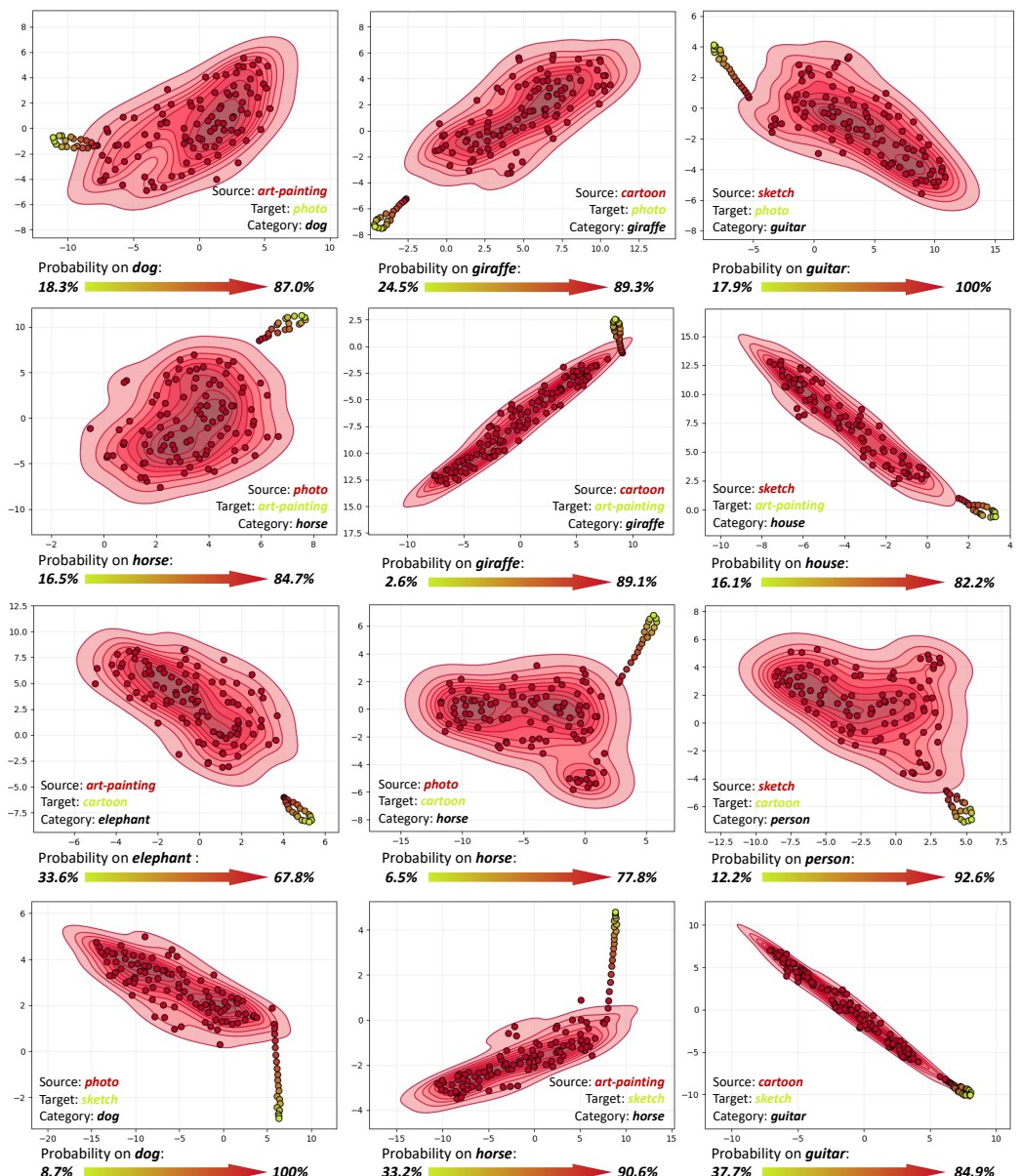

Figure 6: **More visualizations of the iterative adaptation on PACS.** We visualize the adaptation of samples from different target domains to source domains on PACS. Each subfigure shows the adaptation of one target sample to one source domain. The adaptation procedure of the target samples is represented by the gradient color from green to red. In each figure, the target sample has the same label as the source data, but is mispredicted due to domain shifts. During adaptation, the target samples gradually approach the source distributions and eventually get correct predictions.

(10). Due to this supervision, the model is trained to learn to adapt out-of-distribution samples to the source distribution while being able to maintain the correct categorical information conditioned on $\mathbf{z}$ Although trained only on source domains, the ability can be generalized to the target domain since it is trained by mimicking different domain shifts during training. To further show that $\mathbf{z}$ captures the categorical information in $\mathbf{x}$, we visualized the features $\mathbf{x}_t$ and latent variables $\mathbf{z}_t$ of the target samples in Figure 8, which shows that $\mathbf{z}_t$ actually captures the categorical information. Moreover, $\mathbf{z}_t$ is more discriminative than $\mathbf{x}_t$ as shown in the figure. Although $\mathbf{z}_t$ is approximated by only $\mathbf{x}_t$ during inference.

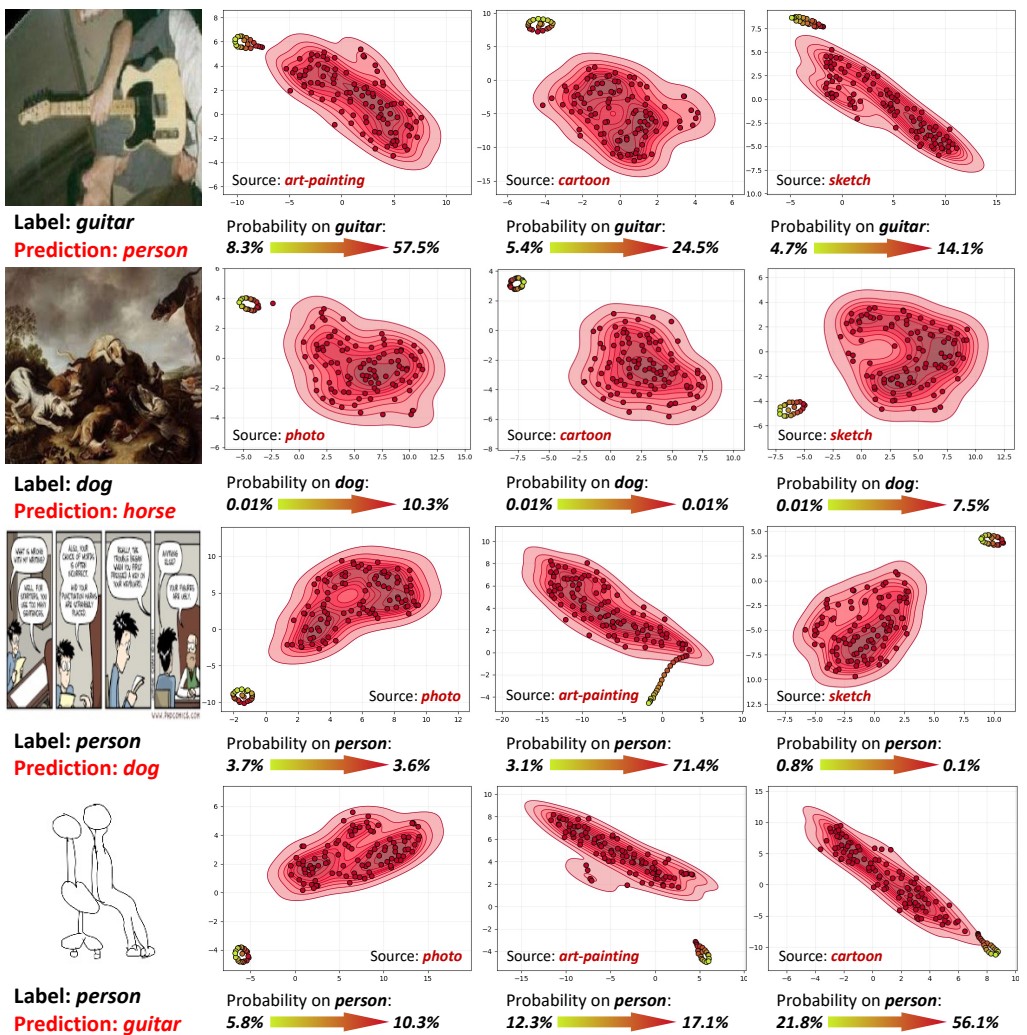

Figure 7: **Failure case visualizations of our method on PACS.** The visualization settings are the same as Figure 6. Our method makes wrong predictions on samples with complex background or multiple objects. However, our method still achieves good adaptation of these target samples to some source domains, which shows its effectiveness.

Moreover, the categorical latent variable benefits the correctness of the model in the case that the target samples are adapted to previously unexplored regions with very large numbers of steps. Our optimization objective is to minimize the energy to adapt the sample, therefore it is possible that the energy of the target samples is lower than the source data after very large numbers of steps. In this case, the adapted samples could arrive in previously unexplored regions due to the limit of source data. This can further be demonstrated in Figure 4, where the performance of the adapted samples drops after large numbers of steps, reaching a low energy value. Additionally, in the unexplored regions, the classifier could not be well trained, which might also be a reason for causing the performance drop. This is also one reason that we set the number of steps as a small value, e.g., 20 and 50. The categorical latent variable benefits the correctness of the model in such cases as also can be found in Figure 4. The oracle model shows almost no performance degradation even with small energy values after adaptation. The model with the latent variable $p(\mathbf{z}|\mathbf{x})$ is also more robust to the step numbers and energy values than the model without $\mathbf{z}$. These results show the role of the latent variable in preserving the categorical information during adaptation and somewhat correcting prediction after adaptation.

With the categorical latent variable $\mathbf{z}$, it is natural to make the final prediction directly by $p_\phi(\mathbf{y}|\mathbf{z})$ without the sample adaptation procedure. However, here we would like to clarify that it is sub-optimal.

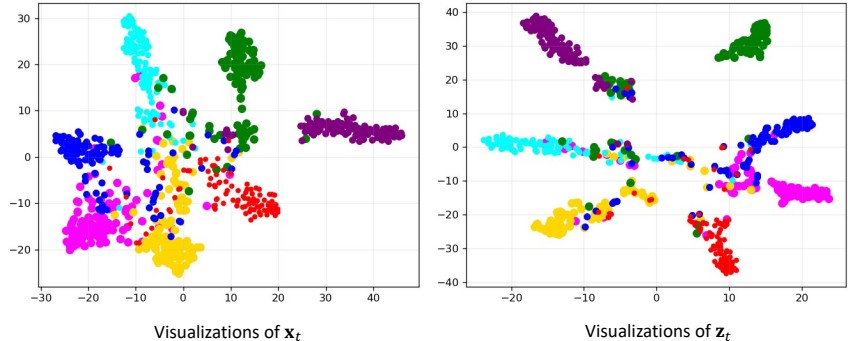

Figure 8: **Visualizations of the target features $x_t$ and categorical latent variables $z_t$.** We use *art-painting* on PACS as the target domain. Different colors denote different categories. $z_t$ is obtained by $p_\phi(z_t|x_t)$. Most data points of $z_t$ are clustered according to their labels, demonstrating that $z_t$ can capture the categorical information.

Table 4: **Analyses on the categorical latent variable.** As expected, prediction on the adapted target samples $x$ performs better than prediction directly on the categorical latent variable $z$.

(a) Overall comparisons on PACS and Office-Home.

|  | PACS | | Office-Home | |
| --- | --- | --- | --- | --- |
|  | ResNet-18 | ResNet-50 | ResNet-18 | ResNet-50 |
| predict directly on $z$ | 82.46 $\pm 0.34$ | 85.95 $\pm 0.33$ | 64.49 $\pm 0.25$ | 70.60 $\pm 0.53$ |
| predict on adapted target samples $x$ | **84.92** $\pm 0.59$ | **87.70** $\pm 0.28$ | **66.31** $\pm 0.21$ | **72.07** $\pm 0.38$ |

(b) Detailed comparisons on PACS.

|  | Photo | Art-painting | Cartoon | Sketch | *Mean* |
| --- | --- | --- | --- | --- | --- |
| **Predict directly on $z$** | | | | | |
| No adaptation | 94,22 $\pm 0.25$ | 79.52 $\pm 0.21$ | 80.46 $\pm 0.43$ | 75.63 $\pm 0.68$ | 82.46 $\pm 0.34$ |
| **Predict on $z$ with model adaptation (Tent)** | | | | | |
| Adaptation with 1 sample per step | 80.49 $\pm 0.27$ | 44.14 $\pm 0.38$ | 51.49 $\pm 0.44$ | 30.28 $\pm 0.66$ | 51.60 $\pm 0.37$ |
| Adaptation with 16 samples per step | 93.65 $\pm 0.33$ | 80.20 $\pm 0.24$ | 76.90 $\pm 0.52$ | 68.49 $\pm 0.72$ | 79.81 $\pm 0.31$ |
| Adaptation with 64 samples per step | 96.04 $\pm 0.33$ | 81.91 $\pm 0.37$ | 80.81 $\pm 0.64$ | 76.33 $\pm 0.65$ | 83.77 $\pm 0.41$ |
| Adaptation with 128 samples per step | **97.25** $\pm 0.24$ | **84.91** $\pm 0.31$ | 81.12 $\pm 0.47$ | 76.80 $\pm 0.83$ | **85.02** $\pm 0.49$ |
| **Predict on adapted target samples $x$ with our method** | | | | | |
| Adaptation with 1 sample $x$ | 96.05 $\pm 0.37$ | 82.28 $\pm 0.31$ | **81.55** $\pm 0.65$ | **79.81** $\pm 0.41$ | **84.92** $\pm 0.59$ |

The latent variable is dedicated to preserving the categorical information in $x$ during adaptation. It still contains the domain information of the target samples. Therefore, it is not optimal to directly make predictions on the latent variable $z$ due to the domain shifts between the $z$ and the source-trained classifiers. By contrast, the proposed method moves the target features close to the source distributions to address domain shifts while preserving the categorical information during adaptation. To show how it works by direct prediction on $z$, we provide the experimental results of making predictions only from $z$ in Table 4. As expected it is worse than predictions on the adapted target features $x$, demonstrating the analysis we provided above.

To show the advantages of our method, we also combine the prediction of the latent variable $z$ with model adaptation methods. We use the online adaptation proposed by Wang et al. (2021), where all target samples are utilized to adapt the source-trained models in an online manner. The model keeps updating step by step. In each step, the model is adapted to one batch of target samples. As shown in Table 4b, with large numbers of target samples per step, e.g., 128, the adaptation with Tent is competitive. However, when the number of samples for online adaptation is small, e.g., 1 and 16, the performance of the adapted model even drops, especially for single sample adaptation. By contrast, our method adapts each target sample to the source distribution. All target samples are adapted and

Table 5: **Training cost of the proposed method.** Compared with ERM, our method has about 20% more parameters, most of which come from the energy functions of source domains. Similar to test time, the time cost of training increases along with the number of steps of the energy-based model.

|  | Parameters | Adaptation steps | 10000 iterations training time |
|---|---|---|---|
| ERM | 11.18M | - | 6.2 h |
| *This paper* | 13.73M | 20 | 7.9 h |
|  |  | 40 | 9.4 h |
|  |  | 60 | 10.6 h |
|  |  | 80 | 12.1 h |
|  |  | 100 | 14.0 h |

Table 6: **Comparisons on PACS.** Our method achieves best mean accuracy with a ResNet-18 backbone and is competitive with ResNet-50.

| Backbone | Method | Photo | Art-painting | Cartoon | Sketch | *Mean* |
|---|---|---|---|---|---|---|
| ResNet-18 | Dou et al. (2019) | 94.99 | 80.29 | 77.17 | 71.69 | 81.04 |
|  | Iwasawa & Matsuo (2021) | - | - | - | - | 81.40 |
|  | Zhao et al. (2020) | **96.65** | 80.70 | 76.40 | 71.77 | 81.46 |
|  | Wang et al. (2021) | 95.49 | 81.55 | 77.67 | 77.64 | 83.09 |
|  | Zhou et al. (2020b) | 96.10 | **84.10** | 78.80 | 75.90 | 83.70 |
|  | Xiao et al. (2022) | 95.87 | 82.02 | 79.73 | 78.96 | 84.15 |
|  | *This paper* | 96.05 ±0.37 | 82.28 ±0.31 | **81.55** ±0.65 | **79.81** ±0.41 | **84.92** ±0.59 |
| ResNet-50 | Dou et al. (2019) | 95.01 | 82.89 | 80.49 | 72.29 | 82.67 |
|  | Dubey et al. (2021) | - | - | - | - | 84.50 |
|  | Iwasawa & Matsuo (2021) | - | - | - | - | 85.10 |
|  | Zhao et al. (2020) | **98.25** | 87.51 | 79.31 | 76.30 | 85.34 |
|  | Gulrajani & Lopez-Paz (2020) | 97.20 | 84.70 | 80.80 | 79.30 | 85.50 |
|  | Wang et al. (2021) | 97.96 | 86.30 | 82.53 | 78.11 | 86.23 |
|  | Seo et al. (2020) | 95.99 | 87.04 | 80.62 | **82.90** | 86.64 |
|  | Xiao et al. (2022) | 97.88 | **88.09** | 83.83 | 80.21 | **87.51** |
|  | *This paper* | 97.67 ±0.14 | **88.00** ±0.29 | **84.75** ±0.39 | 80.40 ±0.38 | **87.70** ±0.28 |

predicted equally and individually. The overall performance of our method is comparable to Tent with 128 samples per adaptation step.

**Time cost with different adaptation steps.** As the number of steps increases, both the training and test time cost consistently increases for all target domains. Without adaptation, the test time cost for one test batch is about 0.05 second. The 20-step adaptation will take about 0.1 extra second. This number will increase to 0.25 second with 50 steps. The test time increases by more than 0.5 seconds for 100 iterations, which is ten times that without adaptation and might limit the application of the proposal. The training time cost is more than two times that for ERM for 100 iterations as shown in Table 5. Since the extra time cost is mainly caused by the iterative adaptation, potential solutions can be speeding up the Langevin dynamics with some optimization techniques like Nesterov momentum (Nesterov, 1983), or exploring some one-step methods for the target sample adaptation. In other experiments on PACS in the paper we use 20 steps for all target domains considering both the overall performance and the time cost.

**Detailed comparisons.** We provide the detailed performance of each target domain on PACS (Table 6), Office-Home (Table 7), and PHEME (Table 8). On PACS, our method achieves competitive results on each target domain and the best overall performance with both ResNet-18 and ResNet-50 as the backbone. Moreover, our method performs better than most of the recent model adaptation methods (Wang et al., 2021; Dubey et al., 2021; Iwasawa & Matsuo, 2021; Xiao et al., 2022). The conclusion on Office-Home and PHEME is similar to that on PACS. We achieve competitive and even better performance on each target domain.

**Results on single source domain generalization.** To show the ability of our method of handling corruption distribution shifts and single source domain generalization, we conduct some experiments on CIFAR-10-C and ImageNet-C. We train the model on original data and evaluate it on the data with 15 types of corruption.

Table 7: **Comparisons on Office-Home.** Our method achieves the best mean accuracy using both a ResNet-18 and ResNet-50 backbone.

| Backbone | Method | Art | Clipart | Product | Real World | *Mean* |
|---|---|---|---|---|---|---|
| ResNet-18 | Iwasawa & Matsuo (2021) | 47.00 | 46.80 | 68.00 | 66.10 | 57.00 |
| | Wang et al. (2021) | 56.45 | 52.06 | 73.19 | 74.82 | 64.13 |
| | Xiao et al. (2022)] | 59.39 | **53.94** | **74.68** | 76.07 | 66.02 |
| | *This paper* | **60.08** $_{\pm0.33}$ | 53.93 $_{\pm0.34}$ | 74.50 $_{\pm0.39}$ | **76.74** $_{\pm0.24}$ | **66.31** $_{\pm0.21}$ |
| ResNet-50 | Gulrajani & Lopez-Paz (2020) | 61.30 | 52.40 | 75.80 | 76.60 | 66.50 |
| | Wang et al. (2021) | 62.12 | 56.65 | 75.61 | 77.58 | 67.99 |
| | Dubey et al. (2021) | - | - | - | - | 68.90 |
| | Xiao et al. (2022) | 67.21 | 57.97 | 78.61 | 80.47 | 71.07 |
| | *This paper* | **69.33** $_{\pm0.14}$ | **58.37** $_{\pm0.30}$ | **79.29** $_{\pm0.32}$ | **81.26** $_{\pm0.26}$ | **72.07** $_{\pm0.38}$ |

Table 8: **Generalization beyond image data.** Rumour detection on the PHEME microblog dataset. Our method achieves the best overall performance and is competitive in each domain.

| | Charlie Hebdo | Ferguson | German Wings | Ottawa Shooting | Sydney Siege | *Mean* |
|---|---|---|---|---|---|---|
| ERM Baseline | 79.4 $_{\pm0.25}$ | 77.1 $_{\pm0.36}$ | **75.7** $_{\pm0.12}$ | 68.2 $_{\pm0.48}$ | 75.0 $_{\pm0.28}$ | 75.1 $_{\pm0.29}$ |
| Wang et al. (2021) | 80.1 $_{\pm0.18}$ | 76.9 $_{\pm0.56}$ | 74.7 $_{\pm0.52}$ | **72.0** $_{\pm0.48}$ | 75.4 $_{\pm0.34}$ | 75.8 $_{\pm0.23}$ |
| Xiao et al. (2022) | 81.0 $_{\pm0.52}$ | 77.2 $_{\pm0.25}$ | **75.7** $_{\pm0.31}$ | 70.0 $_{\pm0.46}$ | **76.5** $_{\pm0.22}$ | 76.1 $_{\pm0.21}$ |
| *This paper* | **81.8** $_{\pm0.43}$ | **77.8** $_{\pm0.37}$ | 75.3 $_{\pm0.14}$ | 71.9 $_{\pm0.28}$ | 75.6 $_{\pm0.15}$ | **76.5** $_{\pm0.18}$ |

Table 9: **Experiments on single-source domain generalization.** The model is trained on original data and evaluated on 15 different types of corruption. Our method is competitive with Sun et al. (2020), Rusak et al. (2020) and Hendrycks et al. (2020), and is outperformed by Wang et al. (2021). Mimicking good domain shifts during training is important for our method.

| Method | CIFAR-10-C | ImageNet-C |
|---|---|---|
| Yun et al. (2019) | 31.1 | - |
| Guo et al. (2019) | 25.8 | - |
| Hendrycks et al. (2020) | 17.4 | 51.7 |
| Rusak et al. (2020) | - | 50.2 |
| Sun et al. (2020) | 17.5 | - |
| Wang et al. (2021) | **14.3** | **44.0** |
| *This paper* (noisy data as negative samples) | 21.5 | 55.8 |
| *This paper* (corrupted data as negative samples) | 17.0 | 51.1 |

Since our method needs to mimic distribution shifts to train the discriminative energy-based model during training, for the single source domain setting, we generate the negative samples by adding random noise to the image and features of the clean data. We also use the other 4 corruption types (not contained in the evaluation corruption types) as the negative samples during training, which we regard as "corrupted data as negative samples". Note that these corrupted data are only used as negative samples to train the energy-based model. As shown in Table 9, by mimicking better domain shifts during training, our method achieves competitive results with Sun et al. (2020). We also compare our method with some data-augmentation-based methods (e.g., Mixup (Guo et al., 2019), CutMix (Yun et al., 2019) and AugMix (Hendrycks et al., 2020)), our sample adaptation is also competitive with these methods. The proposed method performs worse with a single source domain, although we generate extra negative samples to mimic the domain shifts. The reason can be that the randomly generated domain shifts do not well simulate the domain shift at test time.

**Analyses for ensemble prediction.** We conduct several experiments on PACS to analyze the ensemble inference in our method. We first provide the results of each source-domain-specific classifier before and after sample adaptation. As shown in Table 10, Although the performance before and after adaptation to different source domains are different due to domain shifts, the proposed sample adaptation to most of the source domains performs better. Moreover, the ensemble inference further improves the overall performance of both without and with sample adaptation, where the results with sample adaptation are still better, as expected.

We also try different aggregation methods to make the final predictions. The results are provided in Table 11. The best results in Table 10 are comparable, but it is difficult to find the best source

Table 10: **Sample adaptation to each source domain on PACS.** The experiments are conducted on ResNet-18. Due to the domain shifts between the target domain and different source domains, the performance before and after adaptation are different. The results with sample adaptation to most of the source domains are better than those without adaptation. The ensemble inference further improves the overall performance, where the results with sample adaptation are still better than that without adaptation.

(a) Photo

|  | Art-painting | Cartoon | Sketch | *Ensemble* |
|---|---|---|---|---|
| w/o adaptation | 95.79 ±0.23 | 95.03 ±0.27 | 95.05 ±0.42 | 95.12 ±0.41 |
| w/ adaptation | 95.81 ±0.27 | 94.69 ±0.21 | 95.99 ±0.45 | 96.05 ±0.37 |

(b) Art-painting

|  | Photo | Cartoon | Sketch | *Ensemble* |
|---|---|---|---|---|
| w/o adaptation | 78.52 ±0.43 | 79.68 ±0.37 | 79.83 ±0.52 | 79.79 ±0.64 |
| w/ adaptation | 81.49 ±0.33 | 82.19 ±0.35 | 80.81 ±0.43 | 82.28 ±0.31 |

(c) Cartoon

|  | Photo | Art-painting | Sketch | *Ensemble* |
|---|---|---|---|---|
| w/o adaptation | 79.05 ±0.33 | 78.93 ±0.41 | 78.80 ±0.55 | 79.15 ±0.37 |
| w/ adaptation | 81.09 ±0.38 | 80.44 ±0.31 | 80.32 ±0.71 | 81.55 ±0.65 |

(d) Sketch

|  | Photo | Art-painting | Cartoon | *Ensemble* |
|---|---|---|---|---|
| w/o adaptation | 78.32 ±0.56 | 76.98 ±0.73 | 76.16 ±0.82 | 79.28 ±0.82 |
| w/ adaptation | 79.72 ±0.43 | 79.69 ±0.67 | 79.77 ±0.45 | 79.81 ±0.41 |

Table 11: **Analyses of different aggregation methods for the predictions.** The experiments are conducted on PACS using ResNet-18. The results with different aggregation methods are similar while ensemble inference performs slightly better.

| Aggregation methods | Photo | Art-painting | Cartoon | Sketch | *Mean* |
|---|---|---|---|---|---|
| Adaptation to the closest source domain | 95.41 ±0.28 | 79.86 ±0.41 | 79.67 ±0.44 | 78,97 ±0.72 | 83.48 ±0.43 |
| Weighted average of adaptation to different source domains | 95.93 ±0.33 | 82.18 ±0.37 | 81.24 ±0.52 | 79.54 ±0.77 | 84.76 ±0.55 |
| Most confident prediction after adaptation | 95.77 ±0.25 | 81.93 ±0.31 | 80.67 ±0.65 | 79.25 ±0.62 | 84.41 ±0.32 |
| Ensemble (This paper) | 96.05 ±0.37 | 82.28 ±0.31 | 81.55 ±0.65 | 79.81 ±0.41 | 84.92 ±0.59 |

domain for adaptation before obtaining the results. We tried to find the closest source domain of each target sample by the cosine similarity of feature representations and the predicted confidence. We also tried to aggregate the predictions by weighted average according to the Cosine similarity. With cosine similarity, the weighted averaged results are comparable to the common ensemble method we used in the paper, but the results of adaptation to the closest source domain are not so good. The reason can be that the cosine measure is not able to estimate the domain relationships, showing that it is difficult to reliably estimate the relationships between source and single target samples. The results obtained by using the most confident adaptation are also not as good as the ensemble method, although comparable. The reason can be that ensemble methods introduce uncertainty into the predictions, which is more robust.

**Benefit for larger domain gaps.** To show the benefit of our proposal for domain generalization scenarios with large gaps, we conduct experiments on rotated MNIST and Fashion MNIST. The results are shown in Figure 9. The models are trained on source domains with rotation angles from $15°$ to $75°$, $30°$ to $60°$, and $30°$ to $45°$, and always tested on target domains with angles of $0°$ and $90°$. As the number of domains seen during training decreases the domain gap between source and target increases, and the performance gaps between our method and others becomes more pronounced. Notably, when comparing our method with the recent test-time adaptation of Xiao et al. (2022), which adapts a model to each target sample, shows adapting target samples better handles larger domain gaps than adapting the model.

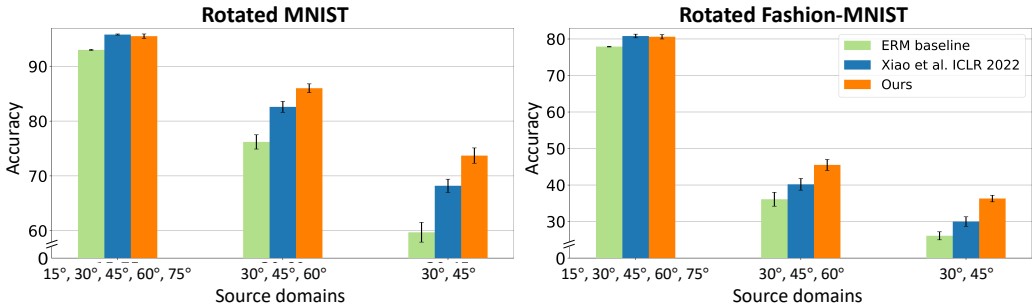

Figure 9: **Benefit for larger domain gaps.** We train on different source set distributions and evaluate on target sets with rotation angles of $0°$ and $90°$. As the domain gap between source and target sets increases, our method performs better than alternatives.

