# OpenReview forum: "Energy-Based Test Sample Adaptation for Domain Generalization"
_ICLR.cc/2023/Conference — ICLR 2023 poster_

### Official Review · Reviewer_Y7Jd · 2022-10-14

**Confidence:** 4
**Correctness:** 2
**Technical Novelty And Significance:** 3
**Empirical Novelty And Significance:** 3
**Recommendation:** 6

**Clarity, Quality, Novelty And Reproducibility:**

As discussed above, sufficient novelty is contained in the proposed method. Clarity and reproducibility lack sometimes, among others for the following reasons (and the one mentioned above under weaknesses):
* Several important details are missing such as: dimensionality of z; how are heads for predicting z, y and the energy defined; what is N in equation 11? In the current form, the proposed method is hard to reproduce.
* The textual discussion after Equation 7 is lengthy and not always correct. It would benefit from another iteration of shortening and making the discussion more precise. Example "The fourth term minimizes the energy and maximizes the prediction log-likelihood of the samples from the model distribution." -> because of the stop-gradient, this term does not optimize energy/log-likelihood of a given sample, but rather increase the probability of such low-energy/high likelihood samples under $q$.
* Algorithm 1 states that contrastive samples at training time come with probability 50% from some set $\mathcal{B}$. This is not discussed in the main text and also not motivated
* Initialization and update of $d_{x^j}$ remains unclear.

Minor points:
 * In Figure 5, why is the energy without conditioning on z higher than when conditioning? It seems that in such an unconstrained setting, a lower energy should be feasible.

**Strength And Weaknesses:**

Strengths:
 * Using energy-based density modelling as basis for test-time adaptation and adapting the sample rather than the model are novel ideas (to my knowledge).
 * The presentation of background and method in Section 2 is thorough and complete.
 * The proposed method is well motivated and clearly derived from first principles.
 * Useful illustrations are provided that illustrate essential parts of the problem/method
 * The empirical evaluation is thorough and conducted on diverse benchmarks, including non-image data

Weaknesses:
* The main question for me is if conditioning on the categorical information $z$ in energy-based density modelling is reasonable. This design choice is motivated by "In order to maintain the category information of the target samples during adaptation and promote better classification performance, we further introduce a categorical latent variable in our energy-based model." (Introduction). However, when used within a discriminative model, wouldn't it be more natural to predict classes directly upon this categorical latent variable rather than (i) first adapting a sample to become more likely under the energy with this categorical variable and then (ii) predict based upon this sample? Directly predicting upon the latent variable would be more computationally efficient and it remains unclear why the two step approach (i) + (ii) should result in better predictive performance? Specifically, when making $p_{\phi_i} (y\vert z^n, x)$ unconditional of $x$, that is: $p_{\phi_i} (y\vert z^n)$, how would this perform at inference time?
* The paper somehow (probably unintentional) obfuscate that adaptation is conducted in feature space and not in input space. It is briefly mentioned at the end of Section 2, but is easily missed (I missed it did during the first read). Since this is clearly an essential part of the method's design, it should be presented earlier and clearer. This also applies to Algorithm 1 where both input and its feature representation are denoted by $\mathbf{x}$.

**Summary Of The Paper:**

The paper studies the setting of homogeneous domain generalization, and proposes a novel single-sample based adaptation scheme. In this scheme, the discriminative classifier remains constant but the respective test sample gets adapted in feature space based upon an energy-based objective. The main contribution of the paper is to adjust energy-based density modelling such that it can be employed at test-time for domain adaptation. For this, the energy-based modelling is extended to be conditioned on categorical information inferred from the test sample before adaptation such that test sample adaptation  does not loose categorical information.

**Summary Of The Review:**

In summary, the paper proposes an interesting approach to a relevant problem. Clarity and reproducibility could be improved with a minor revision. At this point, my main concern is whether energy-based sample adaptation as proposed is actually helpful if it is based upon inferred categorical information (which kind of requires solving the classification problem already _before_ the adaptation). This point requires clarification and some solid empirical support before warranting acceptance of this paper.

### Updated after author feedback ###
The authors have addressed my main criticism and have provided preliminary evidence that energy-based test sample adaptation is preferable over standard adaptation in a small batch setting. I am increasing my overall score, assuming that the authors will extend and revise their manuscript accordingly.

---

> ### Author Response · Authors · 2022-11-18
> **Response to Reviewer Y7Jd (2/2)**
>
> **Clarity, Quality, Novelty And Reproducibility**
>
> **Details of the model architecture and hyperparameters**
>
> We regret that these important details were not included in the initial submission and provide them all in Appendix B of the revised paper, as follows:
> * The dimension of $\mathbf{z}$ is the same as the output of the feature extractor $\mathbf{x}$, e.g., 512 for ResNet-18 and 2048 for ResNet-50.
> * To predict $\mathbf{z}$ we use an MLP containing four fully connected layers with ReLU as the activation function. The last layer of the MLP outputs both the mean and standard derivation of the distribution $p(\mathbf{z}|\mathbf{x})$ for further Monte Carlo sampling. The network is shared to generate the posterior distribution $q(\mathbf{z}|\mathbf{d}_\mathbf{x})$.
> * To predict $\mathbf{y}$, we just use one fully connected layer as the classifier.
> * The neural network for the energy function is also an MLP with three fully connected layers and a swish activation function. Since it takes the concatenation of the feature $\mathbf{x}$ and latent variable $\mathbf{z}$ as the input,  the input dimension doubles the output feature of the backbone, i.e., 1024 for ResNet-18 and 4096 for ResNet-50. The final output of the EBM is a scalar, which is processed by a sigmoid function to bound the energy to [0,1].
> * $N$ in equation 11 is 10 for PACS and PHEME, and 5 for Office-Home and DomainNet.
> We will further release all source code and models on our project website to assure reproducibility.
>
> **Revised discussions of Equation 7**
>
> We revised the discussion as follows:
>
> “Different from the first three terms that directly supervise the model parameters $\theta$ and $\phi$, the last term stops the gradients of the energy function $E_{\theta}$ and classifier $\phi$ while back-propagating the gradients to the adapted samples $q_{\theta,\phi}(\mathbf{x},\mathbf{y})$.
> Because of the stop-gradient, this term does not optimize the energy or log-likelihood of a given sample, but rather increases the probability of such samples with low energy and high log-likelihood under the modeled distribution.”
>
> **Discussion and motivation of the replay buffer $\mathcal{B}$**
>
> Following Du & Mordatch (2019),  $\mathcal{B}$ is a replay buffer that stores the past updated samples from the modeled distribution. By sampling from  $\mathcal{B}$ with 50% probability, we initialize the negative samples with either the sample features from other source domains or the past Langevin dynamics procedure. This increases the number of sampling steps and the sample diversity. We added these to the “Implementation details” in the paper.
>
> **Details of obtaining $\mathbf{d}_{\mathbf{x}^j}$**
>
> The parameters $\mathbf{d}_{\mathbf{x}^j}$  are not trainable in our method.  Therefore it does not need to be initialized and updated.
>
> Instead, we simply obtain $\mathbf{d}_{\mathbf{x}^j}$ per training iteration by the center features of the samples from domain $D^j$, which have the same category as the current sample $\mathbf{x}^j$.
>
> **Analyses of the energy values in Figure 5 (Figure 4 in the revised version)**
> Without conditioning on $\mathbf{z}$, there is no guidance of categorical information during sample adaptation. In this case, the sample can be adapted randomly by the energy-based model, regardless of the categorical information. This can lead to the conflict to adapt the target features to different categories of the source data, slowing down the decline of the energy.
> We updated the paper and included this clarification in Section 3.
>
> Thank you.

---

> ### Author Response · Authors · 2022-11-18
> **Response to Reviewer Y7Jd (1/2)**
>
> **Weakness**
>
> **Clarifications and experiments of prediction on the latent variable**
>
> The latent variable is dedicated to preserving the categorical information in $\mathbf{x}$ during adaptation. It still contains the domain information of the target samples. Therefore, it is not optimal to directly make predictions on the latent variable $\mathbf{z}$ due to the domain shifts between the $\mathbf{z}$ and the source-trained classifiers.
> By contrast, the proposed method moves the target features close to the source distributions to address domain shifts while preserving the categorical information during adaptation.
>
> To show how it works by direct prediction on $\mathbf{z}$, we provide the experimental results of making predictions only from $\mathbf{z}$. As expected it is worse than predictions on the adapted target features $\mathbf{x}$, demonstrating the analysis we provided above.
> We added these discussions to Appendix D.
>
> | Settings                                    | Photo        | Art-painting         | Cartoon     | Sketch      | *Mean*        |
> |---------------------------------------------|--------------|-------------|-------------|-------------|-------------|
> | Predict directly on $\mathbf{z}$             | 94.22 ±0.25  | 79.52 ±0.21 | 80.46 ±0.43 | 75.63 ±0.68 | 82.46±0.34  |
> | Predict on the adapted target sample (Ours) | 96.05 ±0.37  | 82.28 ±0.31 | 81.55 ±0.65 | 79.81 ±0.41 | 84.92 ±0.59 |
>
>
> |                                 | PACS         |              | Office-Home  |             |
> |---------------------------------------------|--------------|--------------|--------------|--------------|
> |                                  | ResNet-18    | ResNet-50    | ResNet-18    | ResNet-50    |
> | Predict directly on $\mathbf{z}$              | 82.46 ±0.34  | 85.95 ±0.33  | 64.49 ±0.25  | 70.60 ±0.53  |
> | Predict on the adapted target sample (Ours) | 85.10 ±0.33  | 88.12 ±0.25  | 66.75 ±0.21  | 72.25 ±0.32  |
>
> **Clarification of conducted energy function on feature space**
>
> The reviewer is right. We revised the paper and presented the feature-based adaptation in both the introduction and the beginning of our method (Section 2.2). We also correct the notations of the input features and input images used in our method as $\mathbf{x}$ and $I$, respectively.

---

> > ### Comment · Reviewer_Y7Jd · 2022-11-21
> > **Feedback**
> >
> > I would like to thank the authors for their feedback and clarifications! To my understanding, "Predict directly on z" is a classifier that does not use any data from the target domain? Because in that case, the gap between "Predict directly on z" and "Predict on the adapted target sample (Ours)" is relatively small and I would expect an approach like "Predict directly on z and adapt using TENT etc." would be highly competitive and conceptually simpler?

---

> > > ### Author Response · Authors · 2022-11-22
> > > **Further response to Reviewer Y7Jd**
> > >
> > > We thank the reviewer for further engagement.
> > >
> > > Indeed, "predict directly on $\mathbf{z}$" does not use any data from the target domain. We will provide the suggested experiment to predict on $\mathbf{z}$ with Tent. **We expect to report it in a day or two.**
> > > Whether it would be ‘conceptually simpler’ depends. During training our approach is more complex, but during inference, we have some advantages. At test time we do not need to fine-tune the model for each specific target domain. Moreover, we adapt each individual target sample to the source-trained model without the requirement of extra target data. By contrast, Tent needs a large number of target samples to fine-tune the source-trained models for maximum effectiveness. We will add this discussion to Section 3.

---

> > > ### Author Response · Authors · 2022-11-23
> > > **Experiments with Tent on z**
> > >
> > > Here we provide the experiments that predict on $\mathbf{z}$ with adaptation using Tent (Wang et al., 2021).
> > >
> > > We use the online adaptation proposed by Wang et al., (2021), where all target samples are utilized to adapt the source-trained models in an online manner. The model keeps updating step by step. In each step, the model is adapted to one batch of target samples.
> > >
> > > As shown in the following table, with large numbers of target samples per step, e.g., 128, the adaptation with Tent is competitive. However, when the number of samples for online adaptation is small, e.g., 1 and 16, the performance of the adapted model even drops, especially for single sample adaptation.
> > > By contrast, our method adapts each target sample to the source distribution. All target samples are adapted and predicted equally and individually. The overall performance of our method is comparable to Tent with 128 samples per adaptation step.
> > >
> > > We will add these results and discussions to the paper.
> > >
> > > |                 | Photo        | Art-painting         | Cartoon     | Sketch      | *Mean*         |
> > > |-----------------------------------------|--------------|-------------|-------------|-------------|--------------|
> > > | *Predict directly on $\mathbf{z}$*       |              |             |             |             |              |
> > > | No adaptation                           | 94.22 ±0.25  | 79.52 ±0.21 | 80.46 ±0.43 | 75.63 ±0.68 | 82.46  ±0.34 |
> > > | *Predict on $\mathbf{z}$ with Tent*       |              |             |             |             |              |
> > > | Online adaptation with 1 sample per step              | 80.49 ±0.27  | 44.14 ±0.38 | 51.49 ±0.44 | 30.28 ±0.66 | 51.60 ±0.37  |
> > > | Online adaptation with 16 samples per step     | 93.65 ±0.33  | 80.20 ±0.24 | 76.90 ±0.52 | 68.49 ±0.72 | 79.81 ±0.31  |
> > > | Online adaptation with 64 samples per step   | 96.04 ±0.33  | 81.91 ±0.37 | 80.81 ±0.64 | 76.33 ±0.65 | 83.77 ±0.41  |
> > > | Online adaptation with 128 samples per step | **97.25** ±0.24  | **84.91** ±0.31 | **81.12** ±0.47 | 76.80 ±0.83 | **85.02** ±0.49  |
> > > | *Predict on $\mathbf{z}$ with our method* |              |             |             |             |              |
> > > | Adaptation with 1 sample                | 96.05 ±0.37  | 82.28 ±0.31 | **81.55** ±0.65 | **79.81** ±0.41 | **84.92** ±0.59  |

---

> > > > ### Comment · Reviewer_Y7Jd · 2022-11-24
> > > > **Reply**
> > > >
> > > > I would like to thank the authors for the additional experiments. It provides preliminary evidence that energy-based test sample adaptation is preferable in a small batch setting. A more thorough analysis would be helpful for the revised manuscripts, including an ablation study on the effect of hyperparameters like learning rates on this finding. I will adapt my review accordingly.

---

### Official Review · Reviewer_aESp · 2022-10-19

**Confidence:** 4
**Correctness:** 3
**Technical Novelty And Significance:** 3
**Empirical Novelty And Significance:** 3
**Recommendation:** 8

**Clarity, Quality, Novelty And Reproducibility:**

It is a well-written paper with thorough motivation for the choices made. However, reproducibility lacks sometimes.

Minor questions:

In Sec. 2.1, The sentence "The objective function in eq. (1) encourages" is not finished.

**Strength And Weaknesses:**

Strength:

1. The paper is well written, the motivations for choices in the method are clear and the method is simple yet effective, ripping the benefits of a discriminative energy-based model and Langevin dynamics.

2. The background of the energy-based model is clear and kindful in Section 2.

3. Extensive experiments show that this method performs quite well in comparison to existing DG methods on both image and text benchmarks. Thorough insights are conducted in experiments.

4. Theoretical validation and qualitative results indicate that energy-based test sample adaptation is a good solution to reduce the domain gap at test time.

Weaknesses:
1. It's a bit unclear why the authors haven't included comparisons or discussions about how this method would perform in source-free DA settings. I expect to see whether this method would work in that context too.

2. During the training phase, the proposed sample adaptation framework simultaneously trains the shared backbone network, the classification model and an extra energy function. In the test phase, the target samples are adapted by Langevin dynamics of the energy function. Does this compare fairly to existing DG methods to only use source data at the training step? Depending on the method but it would be a good discussion to have. As an aside, this concern arises because the proposed framework w/o adaptation has achieved comparable performance (e.g., Table 1 and Table 2).

**Summary Of The Paper:**

The paper proposes a domain generalization method with a new discriminative energy-base model. The proposed idea is novel and effective because they adapt the target samples to source distributions instead of generalizing the model to unseen target samples in most previous methods. Theoretically, the authors provide detailed mathematical derivations and proofs. Experimentally, the authors present extensive empirical results and conduct a thorough ablation study.

**Summary Of The Review:**

Interesting paper on domain generalization at test time that proposes a discriminative energy-based model to adapt target samples to source distributions, and works well on six image and text benchmarks. I have concerns about its viability as a source-free domain adaptation method but would be willing to reevaluate based on the authors' response.

---

> ### Author Response · Authors · 2022-11-18
> **Response to Reviewer aESp**
>
> **1. Discussions and comparisons with source-free domain adaptation settings**
>
> Our sample adaptation is indeed source-free, but our setting is different from source-free domain adaptation. Source-free domain adaptation methods follow the domain adaptation setting to fine-tune the source-trained model by the (unlabeled) target set. By contrast, we do sample adaptation at test time but strictly follow the domain generalization setting. In our method, no target sample is available during the training of the models. At test time, each target sample is adapted to the source domains and predicted by the source-trained model individually, without fine-tuning the models or requiring large amounts of (unlabeled) target data.
>
> Naturally, recent source-free domain adaptation methods e.g., Liang et al. 2020, and Liang et al. 2021, obtain better results on PACS (89.1), and Office-Home (74.6). However, the comparison is not fair as they utilize the entire target dataset for fine-tuning at test time. We added these discussions in related work.
>
> **2. Fairness of the comparison with DG methods**
>
> Yes, the comparisons with existing DG methods are fair since we strictly follow the setting of domain generalization. We only use the source data to train the classification and energy-based models during training. At test time, we do our sample adaptation. We make predictions on each individual target sample by just the source-trained models. We do not fine-tune any model on the target data, nor do we use source data at test time. We further clarified this in Section 3.
>
> **Reproducibility**
>
> We provide more details of the model architecture and implementations in the paper to make the method more reproducible.
>
> **Minor questions**
>
> Fixed in the revised paper.
>
> Thank you.

---

### Official Review · Reviewer_yzZP · 2022-10-21

**Confidence:** 5
**Correctness:** 3
**Technical Novelty And Significance:** 3
**Empirical Novelty And Significance:** 2
**Recommendation:** 6

**Clarity, Quality, Novelty And Reproducibility:**

Clarity: The paper is written nicely in my opinion, however there are several parts of the Appendix that authors should incorporate in the main manuscript to make it easier for the reader to digest the approach: particularly the two Algorithms and Figure 6 with the caption in Suppl. Sec B. Quality & Novelty: The approach appears to be novel with its application to domain generalization. Results are however not pushing the state-of-the-art drastically in all respects. Reproducibility: Sufficient, since the authors provide their implementations. However without precisely the executed runs (or saved models), I believe it can be quite challenging to reproduce the exact results.

**Strength And Weaknesses:**

Strengths: 1) The paper is nicely written and the methodology is well described (considering also the clarifications in the Supplementary). 2) Proposed approach is novel and it shows compatibility to different data structures due to its capability to operate in a latent feature space.

Weaknesses: 1) There are several fundamental assumptions implied in the methodology with regards to the extent of the source and target domains, which should have been clarified/discussed. 2) The proposed model is rather complex in its optimization objective since it involves many parametric components put together, and this may somewhat limit its applicability to a broad range of domain generalization problems (e.g., single-domain generalization). 3) Empirical evaluations show minimal performance gains with the proposed approach with respect to the state-of-the-art.


**Summary Of The Paper:**

The paper proposes a test-time sampling based domain generalization approach based on a discriminative energy-based model formulation. The model performs modification of test-time inputs by sampling towards the source data distribution via Langevin dynamics, while simultaneously preserving label information (by augmenting their model with a categorical latent variable) to maintain the discriminative components of the adapted sample. Experiments are performed on both image and text dataset benchmarks on multi-source domain generalization, demonstrating the effectiveness of all components of their model with several ablation studies.

**Summary Of The Review:**

Generally I find this paper to touch an interesting problem with a novel generative-discriminative modeling based approach. Yet, there are some unclear points that I would like to clarify and I summarize them below:
- The model learns an unbounded energy function. It is not clarified though, at test time how does the method ensure a lower energy bound on this sampling process? To illustrate: a given test target domain sample can(?) in fact be evaluated to have a much lower energy value than observed already at the beginning of the sampling process. This would then "adapt" the sample (for a fixed number of SGLD steps) towards previously unexplored regions by the source domain data? How can the model correct for this and give confident decisions in such cases?
- A somewhat implicit assumption of the model is that the categorical latent representation for the test sample will have high fidelity to the correct class. Can the authors discuss how reasonable is this? In case the target sample is represented strongly within an incorrect source domain class (through the only-source-trained backbone), wouldn't this divert the SGLD sampling process off-track, towards the wrong class?
- Is there a deeper analysis of the ensemble classification approach? How robust does this make the model, in comparison to e.g. adapting the target sample only to the closest source domain? Similarly one could also compare the ensemble approach to decision making with respect to the most confident source domain mapping.
- Experiments show that the proposed method can only work on learning from multiple source domains. Interesting observations are presented in the supplementary (Table 9) in that sense, where the model more or less fails on domain generalization from CIFAR-10 to CIFAR-10-C with hand-crafted sources. How is it on ImageNet-C, did the authors try? Also there many simple data augmentation methods that can achieve very high domain generalization of a vanilla classifier to CIFAR-10-C and ImageNet-C, which one should include in Table 9.
- How sensitive are the final accuracies in Table 2 to the # sampling steps? Can the authors provide some additional rows to Table 2 with 10-steps and 50-steps for instance?
- What was the exact feature dimensionality for the EBM modeling network per dataset? How did the authors define the feature extractor backbone? (e.g., is it a post-ReLU activation before the final classifier layer or so?)
- Minor comment: The first sentence of the second paragraph in Sec 2.1 is unfinished.

---

> ### Author Response · Authors · 2022-11-18
> **Response to Reviewer yzZP (3/3)**
>
>
> **Experiments on ImageNet-C and comparisons with data-augmentation-based methods**
>
> Following the reviewer's suggestion, we added more data augmentation results (e.g., Mixup (Guo et al., 2019), CutMix (Yun et al., 2019), ANT (Rusak et al. 2020), AugMix (Hendrycks et al., 2020)) on CIFAR-10-C as shown in the following table. Our results are at least competitive and sometimes better than these methods.
> We also provided the results on ImageNet-C. The conclusion is similar to that on CIFAR-10-C
> We included these results in Table 9.
>
> | Method                                           | CIFAR-10-C | ImageNet-C |
> |--------------------------------------------------|------------|------------|
> | Guo et al. (2019) (Mixup)                        | 25.8       | -          |
> | Yun et al. (2019) (CutMix)                       | 31.1       | -          |
> | Hendrycks et al. (2020) (AugMix)                 | 17.4       | 51.7       |
> | Rusak et al. (2020) (ANT)                        | -          | 50.2       |
> | Sun et al. (2020)                                | 17.5       | -          |
> | Wang et al. (2021)                               | **14.3**   | **44.0**   |
> | This paper (noisy data as negative samples)      | 21.5       | 55.8       |
> | This paper (corrupted data as negative samples)  | 17.0       | 51.1       |
>
> **Results with 10 and 50 sample adaptation steps**
>
> We provided the results with both 10 and 50 Langevin dynamics steps for all datasets in Table 2. Regardless of the number of steps, our sample adaptation has good improvements on all datasets, with larger steps leading to better performance. Considering the trade-off between computational efficiency and performance, we set the step number as 20 in our paper. We added these results in Table 2.
>
> |                     | PACS                |              | Office-Home  |              | DomainNet    | PHEME        |
> |-----------------------------|------------|------------|------------|------------|------------|------------|
> |                     | ResNet18               | ResNet50   | ResNet18        | ResNet50        | ResNet50        | DistilBERT  |
> | w/o adaptation      | 83.33 ±0.43  | 86.05 ±0.37  | 65.01 ±0.47  | 70.44 ±0.25  | 42.90 ±0.34  | 75.4 ±0.13  |
> | 10 steps adaptation | 84.25 ±0.48  | 87.05 ±0.26  | 65.73 ±0.32  | 71.13 ±0.43  | 43.75 ±0.49  | 76.04 ±0.16 |
> | 20 steps adaptation | 84.92 ±0.59  | 87.70 ±0.28  | 66.31 ±0.21  | 72.07 ±0.38  | 44.66 ±0.51  | 76.50 ±0.18 |
> | 50 steps adaptation | 85.10 ±0.33  | 88.12 ±0.25  | 66.75 ±0.21  | 72.25 ±0.32  | 44.98 ±0.43  | 76.92 ±0.18 |
>
> **Details of the energy-based model and backbone**
>
> The feature extractor backbone is defined as a basic Residual Network without the final fully connected layer (classifier). There is no post-ReLU activation before the final classifier layer.
> The Energy-based model takes the concatenation of the feature of samples $\mathbf{x}$ and the latent variable $\mathbf{z}$ as input. Since $\mathbf{z}$ has the same dimension as $\mathbf{x}$, the dimension of the input for the EBM is double the size of the backbone output feature, i.e., 1024 for ResNet-18 and 4096 for ResNet-50. We process the input feature by three fully connected layers with swish activation functions after the first two layers. The final output of the EBM is a scalar, which is processed by a sigmoid function to bound the energy to [0, 1]
> We clarified these implementation details in the Appendix and will provide the source code and models for better reproducibility.
>
> **Minor comment on the unfinished sentence**
>
> Fixed in the revised paper.
>
> Thank you.

---

> > ### Comment · Reviewer_yzZP · 2022-11-21
> > **Thanks for the clarifications and revisions**
> >
> > Thanks a lot to the authors for the detailed clarifications and their efforts!
> >
> > Analysis on the used ensemble classification looks quite supportive to clarify that aspect. Extended single domain generalization experiments and ablations on the number of Langevin sampling steps are also consistent with the rest of the analyses. Inherent complexity of the optimization objective and the test-time limitations of the model remains as the downside of course, which are though well-discussed in the manuscript. I have only one remaining question to the authors.
> >
> > On unbounded energy function: It was previously not clear in the manuscript that the energy function is bounded (see the original definition in 2.1 which is a mapping to R). Hence I did not assume that there was a use of a sigmoid at the output, as this is not very conventional while training a deep EBM? Why a sigmoid (but not relu), do we need an upper bound to represent the energy of a very-far sample? Is there any related work that implements such an EBM backbone, can the authors please point to these work for consistency?

---

> > > ### Author Response · Authors · 2022-11-22
> > > **Further response to Reviewer yzZP**
> > >
> > > We are glad that the answers and additional experiments address your concerns.
> > >
> > > Regarding our choice for the sigmoid function, the reviewer is right that this is unconventional for training an energy-based model.  We follow the suggestion by Du et al. 2021, who state in their code (https://github.com/openai/ebm_code_release) that “applying sigmoid on energy can improve the stability”. We also tried other functions like the square function or without activation function. We find the energy-based model with the sigmoid function is easier to train. We will add the motivation for the sigmoid function to the implementation details in the paper. We will also clarify in the manuscript that the energy function is bounded.
> > >
> > >
> > > *Yilun Du, Shuang Li, Joshua Tenenbaum, and Igor Mordatch. Improved contrastive divergence training of energy based models. In International Conference on Machine Learning. PMLR, 2021.*

---

> > > > ### Comment · Reviewer_yzZP · 2022-11-23
> > > > **further responses**
> > > >
> > > > I understand now, this appears to be an implementation tweak which turned out very important. I believe without constraining the EBM's output range (even though the theory did not mention this in the paper), the fallback of the discussed test-time adaptation limitations would be very strong.
> > > >
> > > > Follow up regarding the additional results for single source domain generalization:
> > > > - Can the authors elaborate specifications of this analysis: "... for the single source domain setting, we generate the negative samples by adding random noise to the image and features of the clean data. We also use the other 4 corruption types (not contained in the evaluation corruption types) as the negative samples during training..". What is the noise scale considered? Is it a single noise level or multiple noise levels to mimic more than one domain? How many source domains are considered in total, one noise distribution and 4 corruption types independently, adding up to 5? It is not really clear how to replicate these results from Table 10, can we clarify in the appendix?

---

> > > > > ### Author Response · Authors · 2022-11-23
> > > > > **Further response to Reviewer yzZP**
> > > > >
> > > > > We will add the details and discussions in the paper. Regarding the experiments on single source domain generalization, the corruptions we use are 'gaussian_blur', 'saturate', 'spatter', and 'speckle_noise'. Each corruption type mimics a different domain, where within each domain the data is corrupted by multiple levels of noise. Together with the original distribution we thus obtain five source domains. We will add these details to the paper and provide the corresponding code to ease replication of the results.

---

> > > > > > ### Comment · Reviewer_yzZP · 2022-11-23
> > > > > > **presenting performance w/o noise category**
> > > > > >
> > > > > > Thanks for the updates, looking forward to the clarifications to be added.
> > > > > >
> > > > > > Given this setting that the authors just explained, it would be more correct to present average CIFAR10-C and ImageNet-C accuracies without considering the noise corruptions category (i.e., only the average performance on the corruptions from categories digital, weather and blur). Otherwise this noise based manipulation of the source domains provides an unfair(?) advantage to the model, compared to the competitors that the authors list on their paper, such as Mixup, AugMix, etc.

---

> > > > > > > ### Author Response · Authors · 2022-11-23
> > > > > > > **Performance w/o noise category**
> > > > > > >
> > > > > > > We agree with the reviewer that our method profits from more domain diversity during training. We provide a comparison with AugMix by Hendrycks et al. (2020) on CIFAR-10-C when averaged on all corruptions and with corruptions without noise types. When averaged on corruptions without noise types, our results are closer to Hendrycks et al. (2020). We will add these results and discussion to the paper.
> > > > > > >
> > > > > > > |                          | All corruptions  | All corruptions without “Noise” |
> > > > > > > |--------------------------|------------------|---------------------------------|
> > > > > > > | Hendrycks et al. (2020)  | 17.4             | 15.6                            |
> > > > > > > | This paper               | 17.0             | 15.6                            |

---

> ### Author Response · Authors · 2022-11-18
> **Response to Reviewer yzZP (2/3)**
>
> **Reasonability of the assumption that the categorical latent representation for the test sample will have high fidelity to the correct class**
>
> The implicit assumption is guaranteed by the training strategy in our method. We now elaborate on these for you.
>
> We minimize the KL divergence to encourage the prior $p_\phi (\mathbf{z}|\mathbf{x})$ to be close to the variational posterior $q_{\phi}(\mathbf{z} | \mathbf{d}_{\mathbf{x}})$. Here $\mathbf{d}_\mathbf{x}$ is the class prototype containing the categorical information. Hence, we train the inference model $p_\phi(\mathbf{z}|\mathbf{x})$ to learn to extract categorical information from a single sample.
>
> Moreover, we also supervise the sample adaptation procedures by the predicted log-likelihood of the adapted samples (the last term in eq. (10)). The supervision is inherent in the objective function of our discriminative energy-based model as in the derivation of eq. (7) and eq. (10). Due to this supervision, the model is trained to learn to adapt out-of-distribution samples to the source distribution while being able to maintain the correct categorical information conditioned on $\mathbf{z}$.
>
> We added these discussions in the analyses of latent variable $\mathbf{z}$ in Appendix D.
>
> **Deeper analysis of the ensemble method**
>
> We thank the reviewer for sharing the insight. We provide the requested results after adaptation per source domain as well as for different aggregation methods. For most source domains, the results with sample adaptation are better than those without adaptation. Ensemble inference further improves performance. Again results with sample adaptation are better than without.
>
> | target domain: Photo        |            |            |            |            |
> |-----------------------------|------------|------------|------------|------------|
> | source domain               | Art-painting        | Cartoon    | Sketch     | Ensemble   |
> | w/o adaptation              | 95.79±0.23 | 95.03±0.27 | 95.05±0.42 | 95.12±0.41 |
> | w/ adaptation               | 95.81±0.27 | 94.69±0.21 | 95.99±0.45 | 96.05±0.37 |
>
> | target domain: Art-painting |            |            |            |            |
> |-----------------------------|------------|------------|------------|------------|
> | source domain               | Photo      | Cartoon    | Sketch     | Ensemble   |
> | w/o adaptation              | 78.52±0.43 | 79.68±0.37 | 79.83±0.52 | 79.79±0.64 |
> | w/ adaptation               | 81.49±0.33 | 82.19±0.35 | 80.81±0.43 | 82.28±0.31 |
>
> | target domain: Cartoon      |            |            |            |            |
> |-----------------------------|------------|------------|------------|------------|
> | source domain               | Photo      | Art-painting        | Sketch     | Ensemble   |
> | w/o adaptation              | 79.05±0.33 | 78.93±0.41 | 78.80±0.55 | 79.15±0.37 |
> | w/ adaptation               | 81.09±0.38 | 80.44±0.31 | 80.32±0.71 | 81.55±0.65 |
>
> | target domain: Sketch       |            |            |            |            |
> |-----------------------------|------------|------------|------------|------------|
> | source domain               | Photo      | Art-painting        | Cartoon    | Ensemble   |
> | w/o adaptation              | 78.32±0.56 | 76.98±0.73 | 76.16±0.82 | 79.28±0.82 |
> | w/ adaptation               | 79.72±0.43 | 79.69±0.67 | 79.77±0.45 | 79.81±0.41 |
>
> We also varied the aggregation method, using (i) the closest source domain to the target sample in terms of cosine similarity of the feature representations and (ii) decision-making with respect to the most confident source domain adaptation. Both methods are reasonable but do not provide an obvious advantage over our ensemble inference. A reason can be that our ensemble method introduces uncertainty into the predictions, which seems slightly more robust. We added the experiments and analysis on ensemble inference in Appendix E.
>
> | Aggregation method                         | Photo        | Art-painting         | Cartoon     | Sketch      | *Mean*        |
> |--------------------------------------------|--------------|-------------|-------------|-------------|-------------|
> | Adaptation to the closest source domain    | 95.41 ±0.28  | 79.86 ±0.41 | 79.67 ±0.44 | 78.97 ±0.72 | 83.48±0.43  |
> | Most confident prediction after adaptation | 95.77 ±0.25  | 81.93 ±0.31 | 80.67 ±0.65 | 79.25 ±0.62 | 84.41 ±0.32 |
> | Ensemble (This paper)                      | 96.05 ±0.37  | 82.28 ±0.31 | 81.55 ±0.65 | 79.81 ±0.41 | 84.92 ±0.59 |

---

> ### Author Response · Authors · 2022-11-18
> **Response to Reviewer yzZP (1/3)**
>
> **Weakness**
>
> **1) Assumptions of the method**
>
> Learning an energy function that well expresses the source data distribution with categorical information, requires us to mimic domain shifts during training. Therefore, we assume the availability of multiple source domains, which is also a commonly used setting for domain generalization (Li et al., 2017; Dou et al., 2019; Zhou et al., 2021). To assure compatibility among domains we further assume they share the same label space. We further clarified these assumptions in Section 2 of the paper.
>
> **2) Complex optimization objective**
>
> We agree with the reviewer that our optimization objective is complex relative to other model adaptation methods. This is because we target a more challenging scenario: adapting each target sample to the source distributions, not vice versa. As our method does not need to further fine-tune the source-trained model at test time, the training procedure is more involved. This also influences our applicability to the single-domain generalization setting as we need to mimic domain shifts during training, independent of the objective function. Nonetheless, we consider the feasibility of our adaptation tactic of interest and hope to encourage the community towards proposing simpler optimizations. We added the discussions on the optimization objective in our conclusion.
>
> **3) Performance gains with respect to the state-of-the-art**
>
> Most state-of-the-art methods adapt the model rather than the samples, which requires batches of target samples at test time. By contrast, our method is more data efficient at test time, avoiding the problem of data collection per target domain in real-world applications. Despite the data efficiency during inference, our method is still comparable and sometimes better than the state-of-the-art, especially on datasets with more categories, e.g., Office-Home and DomainNet. We will better stress the differences and advantages of our proposal in Section 3.
>
> **Clarity, Quality, Novelty, And Reproducibility:**
>
> **Clarity**
>
> We have moved the Algorithms and Figure 6 to the main manuscript.
>
> **Reproducibility**
>
> We added more details on the model architecture and implementations in the paper to aid reproducibility. We will release all source code and models on our project website.
>
> **Summary Of The Review**
>
> **Unbounded energy function and adaptation to unexplored regions**
>
> The energy function is bounded within [0, 1] because we use a sigmoid function after the output layer. At test time, the energy of the adapted samples will have a lower bound of 0. The adaptation procedure is further constrained by the conditioning categorical latent variable, which will benefit the model for correct predictions. We would like to elaborate on this here and also add these to the paper.
>
> Our optimization objective is to minimize the energy to adapt the sample, therefore it is indeed possible that the energy of the target samples is lower than the source data after very large numbers of steps. We agree that in this case, the adapted samples could arrive in previously unexplored regions due to the limit of source data. This can further be demonstrated in Figure 4 of the revised paper (Figure 5 in the original version), where the performance of the adapted samples drops after large numbers of steps, reaching a low energy value. Additionally, in the unexplored regions, the classifier could not be well trained, which might also be a reason for causing the performance drop. This is also one reason that we set the number of steps as a small value, e.g., 20 and 50.
>
> Moreover, the categorical latent variable benefits the correctness of the model in such cases as also can be found in Figure 4. The oracle model shows almost no performance degradation even with small energy values after adaptation. The model with the latent variable $p(\mathbf{z}|\mathbf{x})$ is also more robust to the step numbers and energy values than the model without $\mathbf{z}$. The results show the role of the latent variable in preserving the categorical information during adaptation and somewhat correcting prediction after adaptation.

---

### Official Review · Reviewer_E7hz · 2022-10-25

**Confidence:** 3
**Correctness:** 4
**Technical Novelty And Significance:** 3
**Empirical Novelty And Significance:** Not applicable
**Recommendation:** 6

**Clarity, Quality, Novelty And Reproducibility:**

The clarity is clear.

One question, in this paper, adaption is also used in training. However, it is unclear how useful for this part. Maybe sample adaptation is already  work well during inference. If this is true, the novelty is limit,  large part of paper content is about training.

**Strength And Weaknesses:**

Strength:

1. The energy-based approach of sample adaption for domain adaption is interesting. The background and proposed method (training and inference) are also clear.

2. The proposed method show the effectiveness on several tasks. And the visualization of the adaptation process shows the effectiveness of the method.

3. The authors also point  out the failure case or limitations of the proposed method with complex background.

Weaknesses:

1. Seems hard to work well in complex tasks. And the proposed rely on good feature representation (the shared backbone model).

2. Somewhat unsure whether is fair during inference.  In this paper, the ensemble inference method is used. The adaption can lead the predication more confidently at each domain classifier.  The sum of log p for each domain domain seems reasonable with sample adaption. it is unclear whether without sample adaption is already good enough with voting method.

3. In Figure 5,  energy and accuracy are shown as the number of step increase. The energy is decreasing. However, there is still some space for energy. What is the sample accuracy with lowest energy?

**Summary Of The Paper:**

In this paper, the authors proposed an energy-based sampling adaptations method for domain generalization. In their method, they adapt the unseen target sample to source-domains at test time. A category latent variable is used to sample update. They show the effectiveness in several classification tasks.

**Summary Of The Review:**

Generally, this is a good paper with clear writing. The story is complete.

---

> ### Author Response · Authors · 2022-11-18
> **Response to Reviewer E7hz**
>
> **1. Complex tasks and rely on good features**
>
> We have indeed not tested our approach beyond the four image and text classification tasks, we will add this to the limitations. We do cover complexity variation in the number of categories, from 2 (PHEME) to 345 (DomainNet). We achieve good performance improvements on all datasets, showing our method is capable of solving classification tasks with a complex number of categories (Table 2). Like other works, our approach benefits from a good feature representation in the shared backbone. As also shown in Table 2, with better feature representations, e.g., extracted by ResNet-50, the adaptation on both PACS and OfficeHome improves over a ResNet-18.
>
> **2. Fairness for comparisons with ensemble**
>
> We regret not making the settings and comparisons clear enough. First, we would like to clarify that the results with and without adaptation in Table 1 and Table 2 are all obtained by the same ensemble method. We train domain-specific classifiers for each source domain. The results without sample adaptation are obtained by predicting the target samples through each source classifier and then averaging the predictions. Hence, we believe our inference setting is fair.
>
> Here we further provide head-to-head comparisons for both with and without adaptation. The following tables show the sample adaptation results per source domain on PACS, further demonstrating the benefit of our proposal. We clarified the settings and comparisons in Section 3 and added the new experiments in Appendix E.
>
> | target domain: Art-painting |            |            |            |            |
> |-----------------------------|------------|------------|------------|------------|
> | source domain               | Photo      | Cartoon    | Sketch     | Ensemble   |
> | w/o adaptation              | 78.52±0.43 | 79.68±0.37 | 79.83±0.52 | 79.79±0.64 |
> | w/ adaptation               | 81.49±0.33 | 82.19±0.35 | 80.81±0.43 | 82.28±0.31 |
>
> | target domain: Cartoon      |            |            |            |            |
> |-----------------------------|------------|------------|------------|------------|
> | source domain               | Photo      | Art-painting        | Sketch     | Ensemble   |
> | w/o adaptation              | 79.05±0.33 | 78.93±0.41 | 78.80±0.55 | 79.15±0.37 |
> | w/ adaptation               | 81.09±0.38 | 80.44±0.31 | 80.32±0.71 | 81.55±0.65 |
>
> | target domain: Photo        |            |            |            |            |
> |-----------------------------|------------|------------|------------|------------|
> | source domain               | Art-painting        | Cartoon    | Sketch     | Ensemble   |
> | w/o adaptation              | 95.79±0.23 | 95.03±0.27 | 95.05±0.42 | 95.12±0.41 |
> | w/ adaptation               | 95.81±0.27 | 94.69±0.21 | 95.99±0.45 | 96.05±0.37 |
>
> | target domain: Sketch       |            |            |            |            |
> |-----------------------------|------------|------------|------------|------------|
> | source domain               | Photo      | Art-painting        | Cartoon    | Ensemble   |
> | w/o adaptation              | 78.32±0.56 | 76.98±0.73 | 76.16±0.82 | 79.28±0.82 |
> | w/ adaptation               | 79.72±0.43 | 79.69±0.67 | 79.77±0.45 | 79.81±0.41 |
>
> **3. Accuracy with lower energy**
>
> To show the performance with lower energy, we further explored the energy and accuracy with larger numbers of steps in Figure 5, i.e., 200, 300, 400, 500, 600.
> We found that the energy tends to converge after 400 Langevin dynamics steps and achieves 0.007 with 600 steps. The accuracy of our method with the latent variable $\mathbf{z}$ also tends to converge, with an average of 81,75% on *art-painting* using 600 adaptation steps.
>
> **Necessity of sample adaptation during training**
>
> We would like to clarify that the sample adaptation procedure is learned by mimicking domain shifts during the training stage. By doing so, the model acquires the ability to adapt each target sample to the source distribution at inference time. Without sample adaptation (by mimicking) during training, we cannot train the energy functions to well model the source domain distributions for sample adaptation. We clarified this at the beginning of Section 2.
>
> Thank you.

---

> > ### Comment · Reviewer_E7hz · 2022-12-05
> > **Thanks for the new update!**
> >
> > Thanks for the clarifications and revisions.  Glad to see these updates!

---

### Author Response · Authors · 2022-11-18
**Summary of changes**

We thank all Reviewers for their insightful reviews, sharp comments, and supportive suggestions. Here, we provide a summary of the main updates in the new version. We also provide a response to each reviewer separately.

Updates in the **main manuscript**:

* Introduction: we clarified that the energy function is deployed on the feature representations.
* Section 2: we clarified the assumptions of our method and the necessity of sample adaptation during training. We updated the discussion of the fourth term of eq. (7). We also moved the Algorithms and illustration figure from the Appendix to Section 2. We further clarified the feature-based energy function and updated the notations.
* Section 3: we clarified the settings of our method with and without adaptation. We added the results of adaptation with a different number of steps in Table 2. We further stress the differences and advantages of our proposal compared with test-time adaptation and source-free adaptation methods. We also added the analysis of different energy curves in Figure 5.
* Section 4: we clarified the comparison of the settings of our method and source-free domain adaptation methods.
* Section 5: we added a discussion on the complexity of the objective function.

Updates in the **Appendix**.

* Appendix B: We added detailed implementations, model architectures, and training procedures. We also moved the details of the datasets here.
* Appendix C: We moved the visualizations from the original paper to this appendix.
* Appendix D: We added the adaptation results per source domain and the analysis of ensemble inference on PACS. We added more analysis and discussions on the categorical latent variable $\mathbf{z}$. We also added more comparisons with data-augmentation-based methods on CIFAR-10-C and ImageNet-C.  We moved the experiments of rotated MNIST here.

---

### Decision · Program_Chairs · 2023-01-20

**Decision:**

Accept: poster

**Justification For Why Not Higher Score:**

- Method is complicated and the ideas may not be as broadly applicable


**Justification For Why Not Lower Score:**

- Novel use of energy-based methods for domain generalization that is well-motivated
- Promising experimental results that show the effectiveness of the method



**Metareview: Summary, Strengths And Weaknesses:**

This paper proposes an energy-based algorithm for domain generalization that adapts tests samples to the source distribution at inference time by mimicking domain shifts during training. Reviewers acknowledged the novelty of the energy-based approach that is also well-motivated, and the experiments showing the effectiveness of the method. There were initial concerns regarding the assumptions required for the method to work and the necessity of various components of the method, but these were sufficiently addressed by the authors in the response with additional results. Other concerns were on the complexity of the method and applicability to various settings. In the end, reviewers are inclined towards acceptance given the novelty of approach and promising results. Authors are encouraged to incorporate any remaining comments/suggestions from the reviews in the final version of the paper.

**Note From Pc:**

if the above contains the word "oral" or "spotlight" please see: "oral" presentation means -> notable-top-5% and "spotlight" means -> notable-top-25%. As stated in our emails, we are disassociating presentation type from AC recommendations